# MafB regulates NLRP3 inflammasome activation by sustaining p62 expression in macrophages

Huachun Cui[1], Sami Banerjee [1], Na Xie[1], Tapan Dey[1], Rui-Ming Liu[1], Yan Y. Sanders[1,2] & Gang Liu [1✉]

Activation of the NLRP3 inflammasome is a two-step process: the priming and the activating. The priming step involves the induction of NLRP3 and pro-IL-1β, while the activating step leads to the full inflammasome activation triggered by a NLRP3 activator. Although mechanisms underlying the NLRP3 inflammasome activation have been increasingly clear, the regulation of this process remains incompletely understood. In this study, we find that LPS and *Pseudomonas aeruginosa* cause a rapid downregulation in MafB transcription in macrophages, which leads to a quick decline in the level of MafB protein because MafB is short-lived and constantly degraded by the ubiquitin/proteasome system. We find that MafB knockdown or knockout markedly enhances the NLRP3, but not the NLRP1, NLRC4, or AIM2, inflammasome activation in macrophages. Conversely, pharmacological induction of MafB diminishes the NLRP3 inflammasome activation. Mechanistically, we find that MafB sustains the expression of p62, a key mediator of autophagy/mitophagy. We find that MafB inhibits mitochondrial damage, and mitochondrial ROS production and DNA cytoplasmic release. Furthermore, we find that myeloid MafB deficient mice demonstrate increased systemic and lung IL-1β production in response to LPS treatment and *P. aeruginosa* infection and deficient lung *P. aeruginosa* clearance in vivo. In conclusion, our study demonstrates that MafB is an important negative regulator of the NLRP3 inflammasome. Our findings suggest that strategies elevating MafB may be effective to treat immune disorders due to excessive activation of the NLRP3 inflammasome.

[1] Division of Pulmonary, Allergy, and Critical Care Medicine, Department of Medicine, University of Alabama at Birmingham, Birmingham, AL 35294, USA.
[2] Department of Microbiology and Molecular Cell Biology, Eastern Virginia Medical School, Norfolk, VA 23501, USA. ✉email: gangliu@uabmc.edu

nflammasome plays an essential role in host defense against microbial pathogens. However, excessive inflammasome activation is often a major cause for many immune disorders. Therefore, inflammasomes should be tightly regulated in order for the immune system to mount an optimum antimicrobial inflammatory response, while preventing deleterious tissue damage[1–6].

At least four pattern-recognition receptors (PRRs) have been shown to assemble the canonical inflammasomes, including the nucleotide-binding oligomerization domain (NOD), leucine-rich repeat (LRR)-containing protein (NLR) family members NLRP1, NLRP3 and NLRC4, as well as the protein absent in melanoma 2 (AIM2)[1,7]. Upon sensing their respective ligands, these PRRs bring Caspase 1 monomers into close proximity, assisted by the adapter ASC (apoptosis-associated speck-like protein containing a CARD), a step that is required for the NLRP3 and AIM2, while not necessary but promotional for the NLRP1 and NLRC4, inflammasomes[1]. This process results in Caspase 1 activation via self-enzymatic cleavage. Activated Caspase 1 proteolytically processes pro-IL-1β and pro-IL-18 into their mature forms and cleaves Gasdermin D, the N-terminal fragment of which inserts into cell membrane, forming a pore structure that leads to a lytic type of cell death named pyroptosis, and allows the release of mature IL-1β and IL-18[3,7–12].

The NLRP3 inflammasome is unique in that it requires a two-step activation process[1,3,4,13,14]. The first step is called priming that is usually initiated by the Toll like receptor (TLR) engagement with a microbial ligand, such as lipopolysaccharide (LPS) of the gram-negative bacteria. The priming step is best known for licensing the cells with induction of NLRP3 and pro-IL-1β, whereas the second step, which is triggered by recognition of a NLRP3 activator, fully induces the inflammasome activation[3,7,13,14]. A wide variety of stimuli are known to activate the NLRP3 inflammasome, including microparticles, ATP, cholesterol and microbial toxins[1,3,4,13,14]. However, it remains to be fully understood how these seemingly unrelated activators accomplish such a feat. In the past decade, mitochondrion has been increasingly believed to be an integration point on which the signaling events activated by the diverse NLRP3 stimuli converge[15,16]. Recent studies established that mitochondrial dysfunction and damage, which usually lead to mitochondrial ROS production and mitochondrial DNA release, are crucial steps to the NLRP3 inflammasome activation induced by a majority of the stimuli[16–20]. Cellular factors that preserve or disrupt mitochondrial homeostasis have been found to dampen or heighten the NLRP3 inflammasome activation[17,19,21,22]. Among those, p62, a key component of the autophagy machinery, has been shown to mediate mitophagic clearance of the damaged mitochondria and thereby specifically mitigating the NLRP3 inflammasome[23,24]. Furthermore, p62 was found to be a transcriptional target of NF-κB, which cultivates a notion that p62 constitutes a negative feedback mechanism to curb excessive NLRP3 inflammasome activation[24]. Despite these findings, additional regulatory elements may still exist and remain to be determined. The need for further mechanistic understanding is underscored by other recent findings that challenged the role of mitochondrial ROS and even mitochondria in the NLRP3 inflammasome activation[25].

MafB belongs to the large MAF transcription factor subfamily that binds to a specific DNA element motif (MARE)[26]. MafB has been shown to control endothelial sprouting[27], osteoclast differentiation[28,29], pancreatic α and β cell differentiation[30,31], podocyte generation[32], and monocytic differentiation[33–36]. MafB was also implicated in immune disorders as it regulates macrophage apoptosis, phagocytosis and the complement system[37–39]. However, its role in innate immune response and inflammasome activation is unclear.

In this study, we found that MafB undergoes rapid proteasomal degradation in LPS-treated macrophages. We showed that MafB selectively regulates the NLRP3 inflammasome activation. Mechanistically, we found that MafB is required for the p62 expression to its fullest level and restricts the NLRP3 activator induced mitochondrial damage, and the resultant mitochondrial ROS production and DNA release. Our study indicates that the degradation of MafB is another important step, additional to the induction of NLRP3 and pro-IL-1β, in the priming stage, which fine-tunes the activation of the NLRP3 inflammasome in macrophages.

## Results

**MafB undergoes a rapid downregulation in response to LPS and *P. aeruginosa*, coinciding with the NLRP3 inflammasome priming in macrophages.** During our systematic studies on macrophage activation, inflammation and inflammasome, we found that the protein level of MafB is rapidly decreased in bone marrow-derived macrophages (BMDMs), thioglycollate elicited peritoneal macrophages (PMs), and the macrophage cell line J774A.1 in response to LPS (Fig. 1a–c). Interestingly, the MafB downregulation perfectly coincided with the upregulation of NLRP3 and pro-IL-1β during the LPS priming of the cells (Fig. 1a–c). We also found that *MafB* mRNA levels undergo a similarly swift decline upon LPS treatment (Fig. 1d, e). To characterize this phenomenon in a disease context, we treated macrophages with heat-killed HKPA that primarily activates TLR2 and found that MafB is also decreased after treatment (Fig. 1f). Together, these data suggest that MafB may play a role in LPS and *P. aeruginosa* induced inflammatory response and the NLRP3 inflammasome priming/activation.

Although the MafB transcription is downregulated precipitously in response to LPS, the parallel rapid decline at its protein level suggests that MafB is an unstable protein. To test this hypothesis, we treated macrophages with the protein translation inhibitor cycloheximide (CHX) and chased the protein levels. As shown in Fig. 1g–i, MafB started to decline soon after protein translation was halted, suggesting that MafB is short-lived and constantly degraded at the steady state. These findings also indicate that the decreased level of the MafB protein during the LPS priming is largely resulted from the inhibition of the MafB transcription.

Degradation of intracellular proteins is mediated by proteasome and/or lysosome. To determine the mechanism by which MafB is degraded, we treated macrophages with either the proteasome inhibitor MG132 or the lysosome inhibitor bafilomycin and found that only MG132, but not bafilomycin, swiftly stabilizes MafB (Fig. 1j). Similarly, only MG132, but not bafilomycin, completely reversed MafB decline at the protein level in LPS-treated macrophages (Fig. 1j). Together, these data suggest that MafB degradation is mediated by proteasome, but not by lysosome. Because proteasome-dependent protein degradation is mostly mediated by ubiquitin conjugation, we examined the role of ubiquitin in this process. We blocked the E1 ubiquitin-activating enzyme with the inhibitor MLN7243 and found that the E1 inhibitor led to MafB accumulation and diminished LPS induced MafB degradation (Fig. 1k). We also found that longer exposure of the MafB blot demonstrated appearance of smear bands from macrophages treated with MG132, which is compatible with a pattern of MafB polyubiquitination (Fig. 1l). To further define if MafB is ubiquitinated, MafB was immunoprecipitated with anti-MafB antibody and blotted with anti-ubiquitin antibody. As shown in Fig. 1l, the precipitated MafB was indeed conjugated with ubiquitin when proteasome activity was blocked in macrophages. Taken together, these findings suggest that MafB undergoes constitutive degradation in a ubiquitin-dependent proteasome-mediated pathway in macrophages.

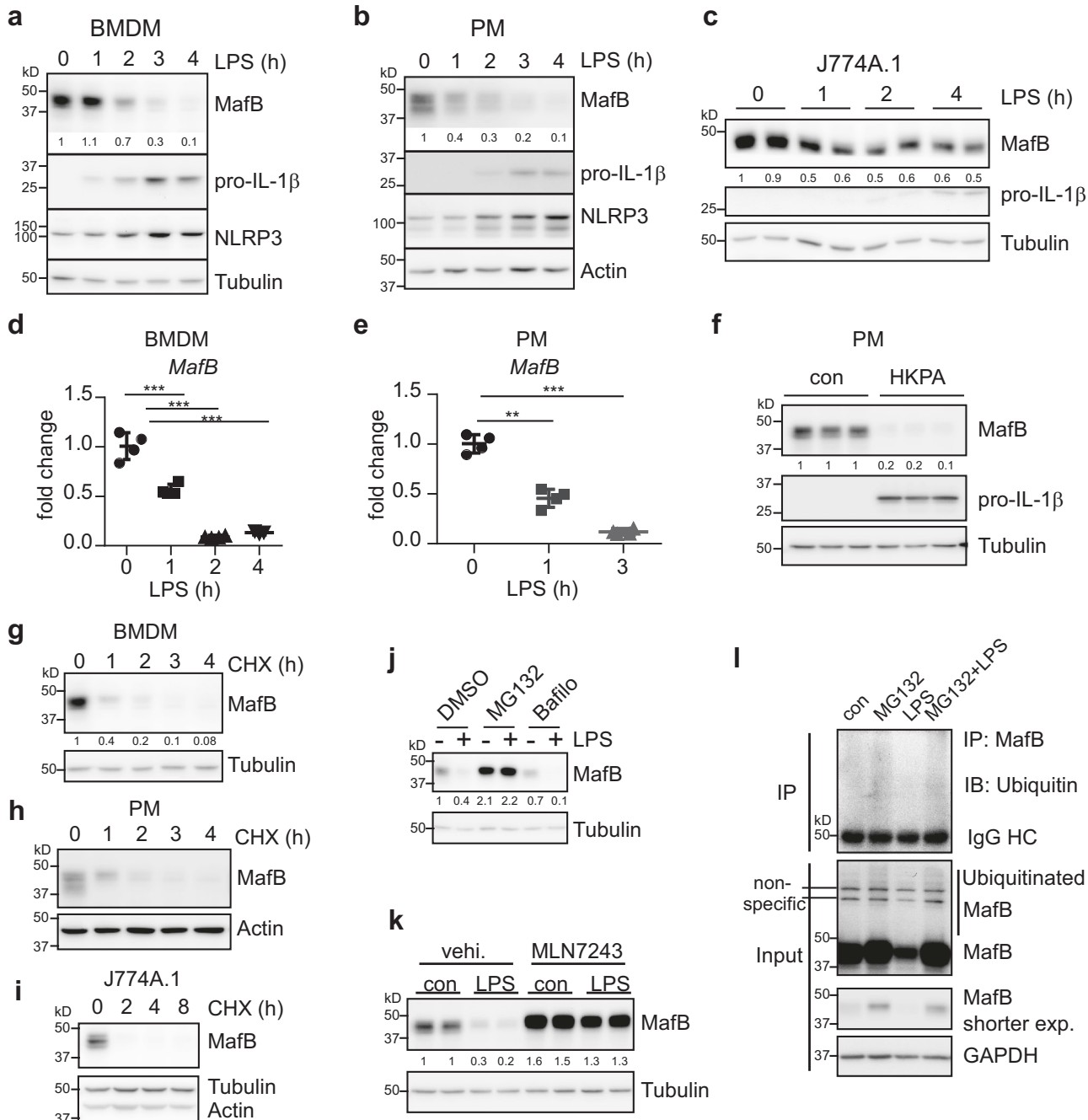

**Fig. 1 MafB undergoes a rapid downregulation in response to LPS and *P. aeruginosa*, coinciding with the NLRP3 inflammasome priming in macrophages.** **a–c** BMDMs, PMs and J774A.1 macrophages were treated with or without 100 ng/ml LPS for the indicated time. Levels of the indicated proteins in the cells were determined by western blotting. Vertically stacked bands in (**a**) originate either from a single membrane or from a replicate membrane, with the same sample loadings. **d, e** BMDMs and PMs were treated with LPS for the indicated time. mRNA levels of *MafB* in the cells were determined by real-time PCR. Representative of three independent experiments. **f** PMs were treated with or without $1 \times 10^7$/ml HKPA for 4 h. Protein levels were determined by western blotting. **g–i** BMDMs, PMs and J774A.1 cells were treated with or without 5 μg/ml CHX for the indicated time. Levels of the indicated proteins were determined by western blotting. **j** BMDMs were pre-treated with 10 μM MG132 or 10 nM Bafilomycin A1 for 30 min, followed by LPS treatment for 3 h. **k** J774A.1 cells were pre-treated with or without 1 μM MLN7243 for 30 min, followed by LPS treatment for 3 h. **l** BMDMs were pre-treated with MG132 for 30 min, followed by treatment of LPS for another 3 h. Cellular extracts were prepared, followed by immunoprecipitation with MafB antibody. Western blotting with the IP and input samples was performed. Mean ± SD; **$P < 0.01$, ***$P < 0.001$. Protein densitometric analyses were performed using ImageJ (NIH) and shown under their corresponding bands. Representative of three independent experiments. HKPA heat-killed *Pseudomonas aeruginosa*, CHX cycloheximide, Bafilo Bafilomycin A1, vehi. vehicle.

**MafB does not participate in the LPS induced early pro-inflammatory response or the expression of NLRP3 and pro-IL-1β.** The rapid decline in MafB in response to LPS indicates that it has a role in the LPS induced pro-inflammatory response and

priming. To test this hypothesis, we knocked down MafB in PMs and BMDMs and treated the cells with LPS, and found that knockdown of MafB has no effect on LPS induced NLRP3 or pro-IL-1β, at either the protein or the transcriptional level (Fig. 2a, b

and Supplementary Fig. 1a). Additionally, MafB knockdown did not affect the expression of pro-inflammatory cytokines, such as TNF-α and IL-6, either (Fig. 2c, d and Supplementary Fig. 1b). To unequivocally demonstrate the role of MafB in LPS induced inflammatory response and priming, we generated myeloid MafB knockout (Mye MafB−/−) mice by crossbreeding the MafB fl/fl (MafB+/+)[26] with the Lyz2 Cre lines. MafB−/− PMs and BMDMs were indeed MafB null (Fig. 2e, f and Supplementary Fig. 1e, f). Importantly, similar to MafB knockdown, there was no difference in LPS induced NLRP3 or pro-IL-1β in between MafB +/+ and MafB−/− macrophages (Fig. 2f, g and Supplementary Fig. 1d, e). MafB knockout did not have effects on other LPS induced pro-inflammatory cytokines or the constitutional levels of the inflammasome adaptor ASC and pro-Caspase 1 (Fig. 2f–h and Supplementary Fig. 1e–g). Taken together, these data suggest that although MafB is rapidly downregulated by LPS, it does not participate in the early inflammatory response to LPS in macrophages.

**MafB knockdown promotes the activation of the NLRP3 inflammasome**. Although MafB did not appear to participate in the

LPS induced NLRP3, pro-IL-1β or the pro-inflammatory cytokines, the coincidence between MafB downregulation and NLRP3 upregulation prompted us to determine if MafB plays a role in inflammasome activation. We started the investigation with the NLRP3 inflammasome. LPS-primed PMs transfected with control or MafB siRNAs were treated with the typical NLRP3 inflammasome activators ATP, Nigericin and monosodium urate crystals (MSU). As shown in Fig. 3a, b, the levels of IL-1β and IL-18 in the cell supernatants from macrophages with MafB knockdown were significantly increased, compared to those from cells transfected with control siRNAs. As inflammasome activation leads to cell pyroptosis[40], we determined if MafB knockdown also affects this process. As shown in Fig. 3c, supernatant levels of lactate dehydrogenase (LDH) were increased in MafB knockdown PMs, indicating more pyroptosis with these cells ensuing the NLRP3 inflammasome activation. To determine that the increased IL-1β in supernatants is not simply caused by an enhanced secretion from MafB knockdown cells, cellular extracts together with supernatants, which captured the entirety of the mature IL-1β, were resolved by SDS-PAGE and blotted with anti-cleaved IL-1β (p17, mature form)

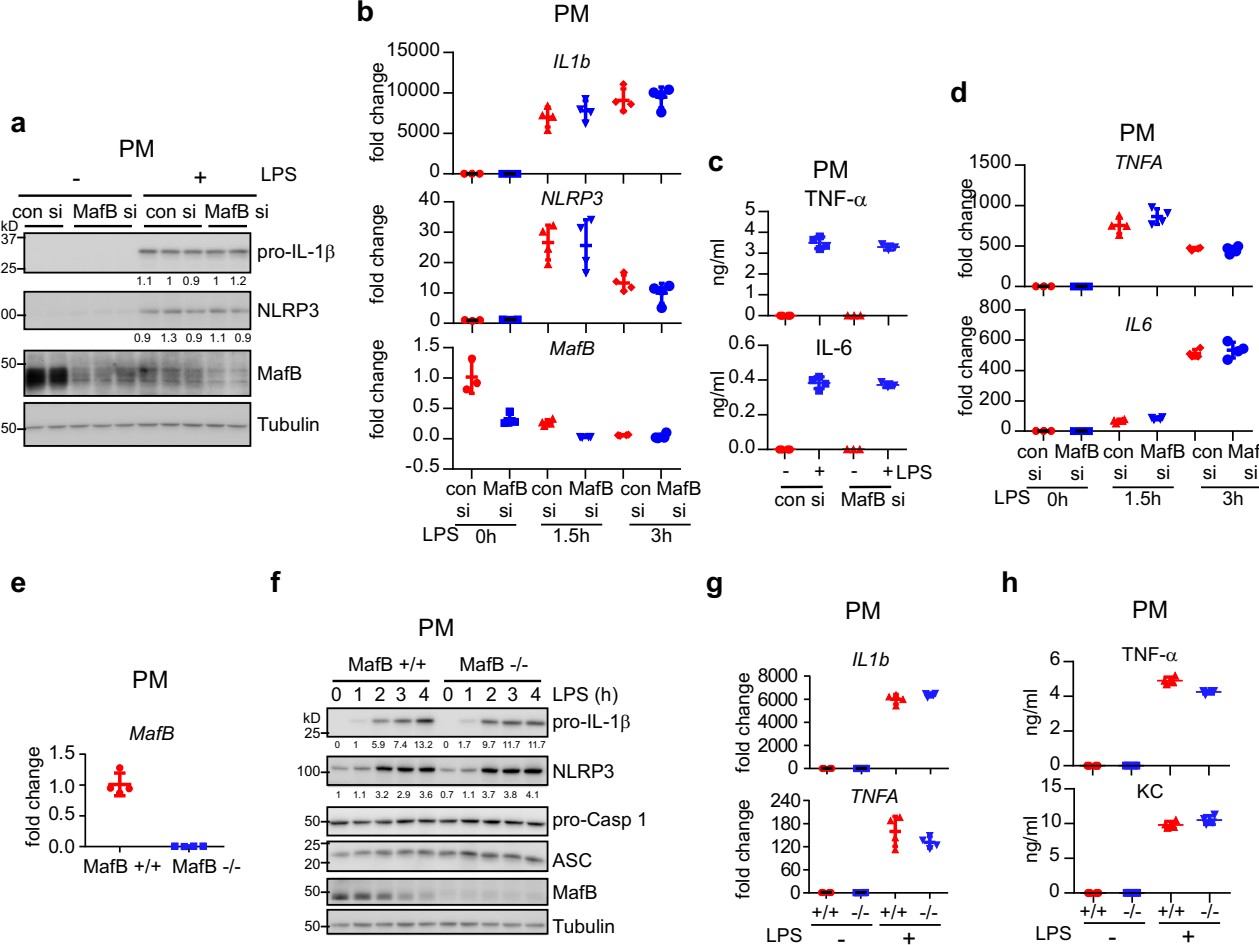

**Fig. 2 MafB does not participate in the LPS induced early pro-inflammatory response or the expression of NLRP3 and pro-IL-1β. a** PMs were transfected with 50 nM control or MafB siRNAs. Forty-eight hours after transfection, cells were treated with or without 100 ng/ml LPS for 3 h. Protein levels were determined by western blotting. Vertically stacked bands originate either from a single membrane or from a replicate membrane, with the same sample loadings. **b** PMs were transfected and treated similarly to (**a**). mRNA levels of the indicated genes were determined by real-time PCR. **c** PMs were transfected and treated as in (**a**). Supernatant TNF-α and IL-6 levels were determined by ELISA. **d** PMs were transfected and treated as in (**b**). The expression of the indicated genes were determined by real-time PCR. **e, f** MafB+/+ and MafB−/− PMs were treated with LPS. *MafB* mRNA levels were determined by real-time PCR (**e**). Protein levels were determined by western blotting (**f**). **g, h** MafB+/+ and MafB−/−. PMs were treated with or without LPS. mRNA levels and supernatant TNF-α and keratinocyte-derived chemokine (KC) were determined by real-time PCR and ELISA, respectively. Mean ± SD. Representative of three independent experiments.

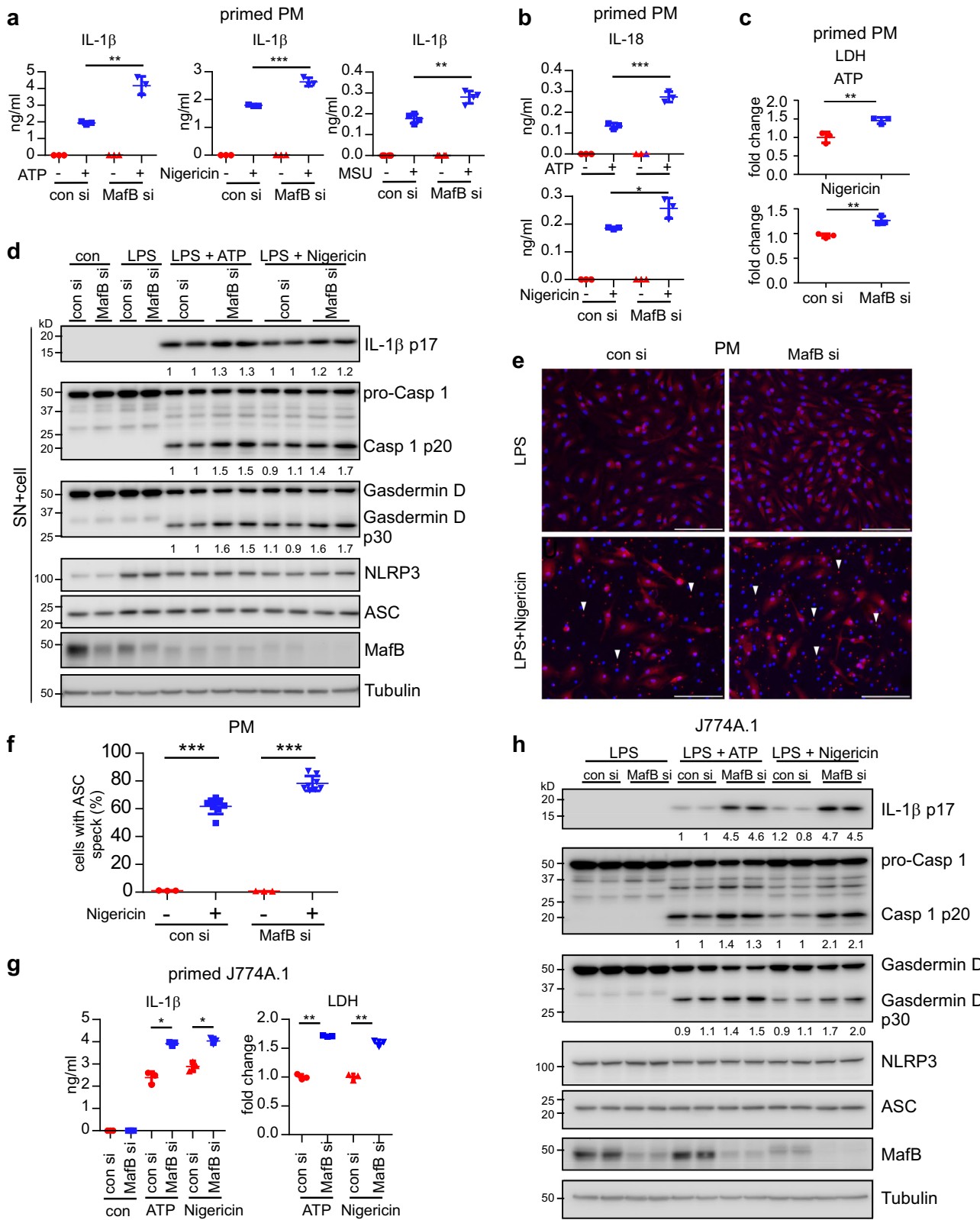

antibody. As shown in Fig. 3d, there was similarly elevated IL-1β p17 in the extract combinations from MafB knockdown PMs. Additionally, there were also more MafB knockdown macrophages with ASC specks than the control cells upon NLRP3 activation (Fig. 3e, f). The MafB downregulation of the NLRP3-dependent IL-1β maturation as well as pyroptosis was again confirmed in J774A.1 macrophages and BMDMs with MafB knockdown (Fig. 3g, h and

Supplementary Fig. 1a, c). Together with the findings that MafB does not affect LPS induced pro-IL-1β or NLRP3, these data suggest that MafB regulates the NLRP3 inflammasome activation.

**MafB−/− macrophages demonstrate enhanced activation of the NLRP3 inflammasome**. To provide further genetic evidence, we examined NLRP3 inflammasome activation in MafB−/−

**Fig. 3 MafB knockdown promotes the activation of the NLRP3 inflammasome. a–c** PMs were transfected with 50 nM control or MafB siRNAs. Forty-eight hours after transfection, cells were treated with 100 ng/ml LPS for 3 h. Cells were then washed and stimulated with or without ATP (5 mM), Nigericin (5 uM) or MSU (1 mM) in serum-free media for an additional 1 h. Levels of the supernatant IL-1β (**a**) and IL-18 (**b**) were determined by ELISA. Levels of the supernatant LDH (**c**) were determined by LDH assay. **d** PMs were transfected, primed and stimulated with the NLRP3 agonists as in (**a**). Supernatant proteins were precipitated by TCA and combined with the respective cellular extracts. Levels of the indicated proteins were determined by western blotting. Vertically stacked bands originate either from a single membrane or from a replicate membrane, with the same sample loadings. **e, f** PMs were transfected with control siRNAs or MafB siRNAs. Forty-eight hours after transfection, cells were primed with LPS for 3 h, followed by stimulation with or without Nigericin for 1 h. Cells were then fixed with 4% PFA and subjected to immunofluorescence staining for ASC. Scale bars: 100 μm (**e**). ASC specks in the cells were quantified (**f**). n = 3–8. **g, h** J774A.1 macrophages were transfected with control siRNAs or MafB siRNAs. 48 h after transfection, cells were primed with LPS for 4 h, followed by stimulation with or without ATP (5 mM) or Nigericin (10 μM) for 1 h. Levels of the supernatant IL-1β and LDH were determined (**g**). Levels of the indicated proteins in the supernatants and cells were determined by western blotting (**h**). Vertically stacked bands originate either from a single membrane or from a replicate membrane, with the same sample loadings. Mean ± SD; *P < 0.05, **P < 0.01, ***P < 0.001. Representative of three independent experiments. LDH lactate dehydrogenase.

macrophages. Similar to the findings with the MafB knockdown, there was an elevated IL-1β in the supernatants of LPS primed MafB−/− PMs and BMDMs treated with ATP or Nigericin, compared to those from MafB+/+ macrophages (Fig. 4a–d). Also concordant with the MafB knockdown, the increased IL-1β in the supernatants was due to the enhanced inflammasome activation, but not a heightened secretion, because there was more mature IL-1β in the combined cellular and supernatant extracts from MafB−/− macrophages (Fig. 4e). Taken together, our siRNA and genetic evidence in both BMDMs, PMs and J774A.1 macrophages firmly establishes an important role for MafB in the NLRP3 inflammasome activation.

**MafB regulates NLRP3 induced ASC oligomerization**. NLRP3 activation triggers its assembly, which leads to ASC recruitment and oligomerization. ASC oligomerization serves as a scaffold, which presents as a macromolecular speck and facilitates pro-Caspase 1 autoproteolysis and the resulted activation. Activated Caspase 1 cleaves pro-IL-1β, pro-IL-18 and Gasdermin D, which then become mature active cytokines and a pore-forming and pyroptosis mediator[3,13,14]. To explore how MafB regulates the NLRP3 inflammasome activation, we assessed ASC oligomerization in MafB+/+ and MafB−/− macrophages. As shown in Fig. 4f, there was markedly increased ASC oligomerization in MafB−/− cells upon NLRP3 activation, compared to MafB+/+ macrophages. Additionally, there were also more MafB−/− macrophages with ASC specks than MafB+/+ cells upon NLRP3 activation (Fig. 4g, h). In accordance with the enhanced ASC oligomerization, Caspase 1 activation was also considerably elevated in MafB−/− macrophages (Fig. 4i, j). Resultantly, Gasdermin D cleavage was elevated in MafB−/− macrophages upon NLRP3 activation (Fig. 4k). These also confirmed the similar findings in MafB knockdown peritoneal and J774A.1 macrophages (Fig. 3d, h). Taken together, these data suggest that MafB plays a role in the NLRP3 activation and the NLRP3-dependent downstream events.

**MafB does not participate in the NLRP1, NLRC4, or AIM2 inflammasome activation**. To determine if MafB participates in other canonical inflammasomes' activation, we treated LPS primed macrophages with the NLRP1 activator muramyl dipeptide (MDP), the NLRC4 activator Flagellin, or the AIM2 activator poly(dA:dT)[1,8]. We found that MafB knockout has no effect on the production of mature IL-1β in macrophages (Fig. 5a, b, d, e). Concordantly, MafB deficiency had no evident effect on ASC oligomerization or pro-Caspase 1 cleavage upon activation of these inflammasomes (Fig. 5b, c). Together, these data suggest that MafB may specifically targets the NLRP3 inflammasome. Of note, the MDP induced mature IL-1β and pro-Caspase 1 cleavage

were barely detectable, suggesting that better and more specific NLRP1 agonists are needed to draw a definitive conclusion.

**MafB restricts the NLRP3 agonists induced mitochondrial damage**. While delineating the mechanism by which MafB specifically regulates the NLRP3 inflammasome, we contemplate an implication of mitochondria in this regulation. Our hypothesis was prompted by the knowledge that mitochondrial ROS generation and mitochondrial DNA release, which are often resulted from mitochondrial damage, are prerequisites to the NLRP3 inflammasome activation[16–20,41]. To test this, we quantitated mitochondrial damage by taking advantage of the distinct features of two MitoTracker fluorescent dyes, MitoTracker Deep Red, of which staining is dependent on mitochondrial potential and enriched in healthy mitochondria, and MitoTracker Green, of which staining is independent on mitochondrial potential and localized in all mitochondria[24]. As shown in Fig. 6a, b, the NLRP3 agonists did induce appearance of damaged mitochondria in primed J774A.1 macrophages, confirming previous reports. Nevertheless, we found that there are more damaged mitochondria in MafB knockdown cells. Consistent with this finding, there were also considerably elevated mitochondrial ROS and mitochondrial DNA in the cytoplasm in MafB knockdown macrophages upon NLRP3 agonists' stimulation (Fig. 6c–e). Together, these data suggest that MafB regulates the NLRP3 inflammasome activation by restricting mitochondria damage and damaged mitochondria-derived ROS and DNA. This notion was reinforced by our finding that the augmented NLRP3 inflammasome in MafB knockdown J774A.1 macrophages was largely abolished when mitochondrial ROS was quenched by MitoTempo in these cells (Fig. 6f, g). Additionally, MafB attenuation of the NLRP3 agonists induced mitochondrial damage, mitochondrial ROS production and DNA release was further confirmed in MafB−/− and MafB knockdown BMDMs and PMs (Supplementary Fig. 2a–i).

**MafB sustains p62 expression in macrophages**. Damaged mitochondria are usually eliminated via mitophagy for cells to return to the mitochondrial homeostasis[17,24,42]. Mitophagic clearance is critically dependent on Parkin mediated ubiquitin conjugation of damaged mitochondria, which is specifically recognized by p62 and other adaptor proteins[24,43]. p62 is a well-established negative regulator of the NLRP3 inflammasome through it promoting mitophagic removal of damaged mitochondria[24,41]. p62 has also been found to be an integral part of the negative feedback regulation of the NF-κB mediated inflammation as p62 is induced late in the pro-inflammatory response in a NF-κB dependent manner[24]. Given that we found MafB restriction of the mitochondria damage in macrophages upon stimulation with the

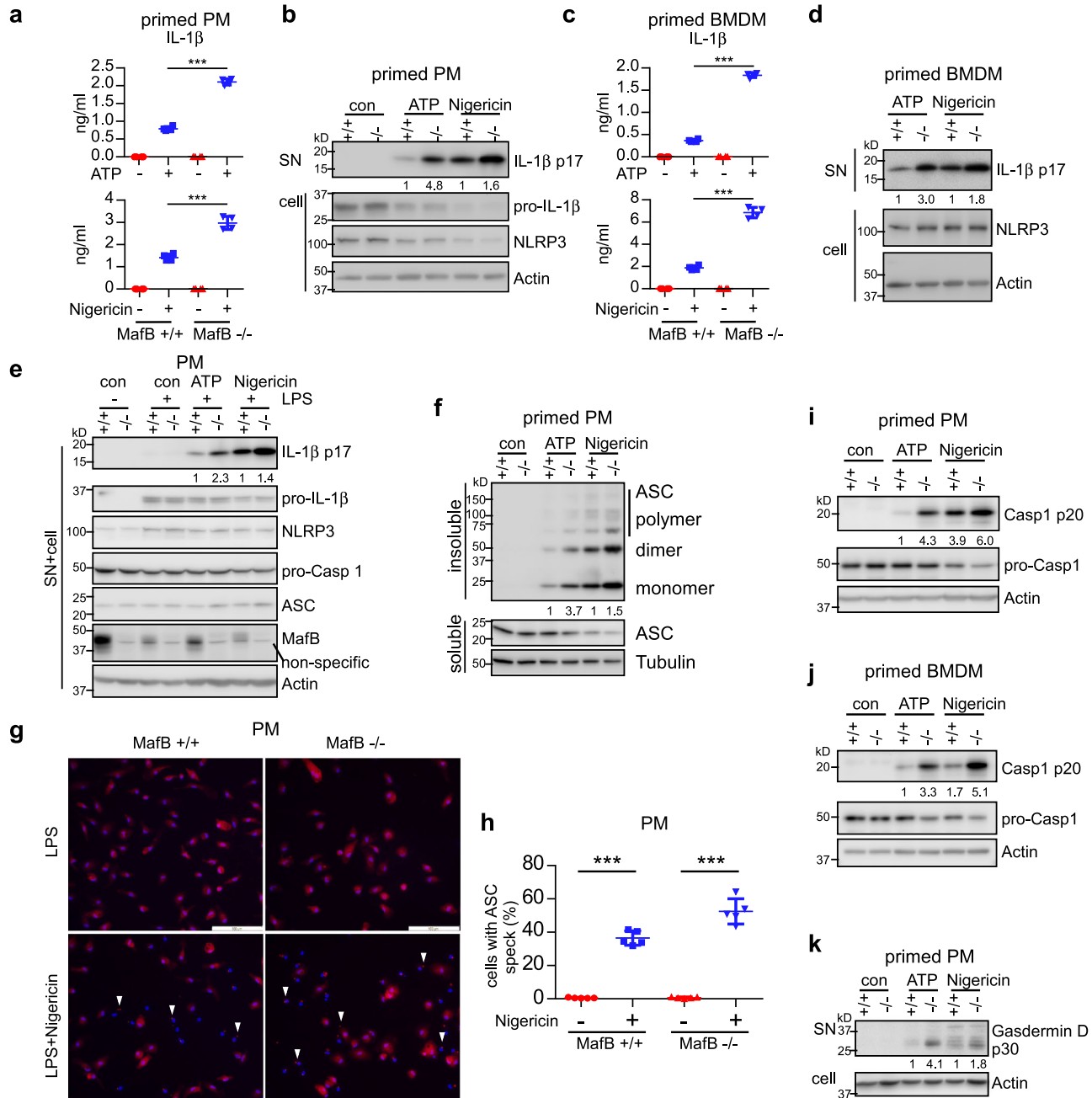

**Fig. 4 MafB −/− macrophages demonstrate enhanced activation of the NLRP3 inflammasome. a–d** Primed MafB+/+ and MafB−/− PMs (**a**, **b**) and BMDMs (**c**, **d**) were washed and stimulated with or without NLRP3 agonists ATP (5 mM) or Nigericin (5 μM) for 1 h. Levels of the supernatant IL-1β were determined by ELISA (**a**, **c**), and levels of the indicated proteins in the cells were determined by western blotting (**b**, **d**). **e** MafB+/+ and MafB−/− PMs were treated as in (**a**). Levels of the indicated proteins in the combined extracts of supernatants and cells were determined by western blotting. Vertically stacked bands originate either from a single membrane or from a replicate membrane, with the same sample loadings. **f** Primed MafB+/+ and MafB−/− PMs were treated as in (**a**). Triton X-100 soluble and insoluble extracts were prepared. Insoluble extracts were cross-linked and ASC oligomerization determined. **g**, **h** Primed MafB+/+ and MafB−/− PMs were stimulated with or without Nigericin for 1 h. Cells were then fixed with 4% PFA and subjected to immunofluorescence staining for ASC. Scale bars: 100 μm (**g**). ASC specks in the cells were quantified (**h**). n = 5. **i–k** Primed MafB+/+ and MafB−/− PMs (**i**, **k**) and BMDMs (**j**) were stimulated with ATP or Nigericin for 1 h. The levels of cleaved Caspase 1 (p20), pro-Caspase 1, and cleaved Gasdermin D (p30) in the supernatants and cells were determined by western blotting. Mean ± SD; **P < 0.01, ***P < 0.001. Representative of three independent experiments.

NLRP3 agonists, we were intrigued to know if this regulation involves p62. To test this hypothesis, we manipulated MafB expression in untreated J774A.1 macrophages and found that p62 level is reduced in cells with MafB knockdown, while components of the NLRP3 inflammasome remained unchanged (Fig. 7a). p62 was induced by LPS, as expected (Fig. 7b). However, the

LPS induction of p62 was diminished in MafB knockdown J774A.1 cells, with or without the NLRP3 stimuli (Fig. 7b). The reduced p62 protein level in the MafB knockdown cells might not be led by a diminished transcription, as there was only a modest decrease of the *p62* transcripts in these cells (Fig. 7c). The MafB dependence of p62 expression was further confirmed in PMs and

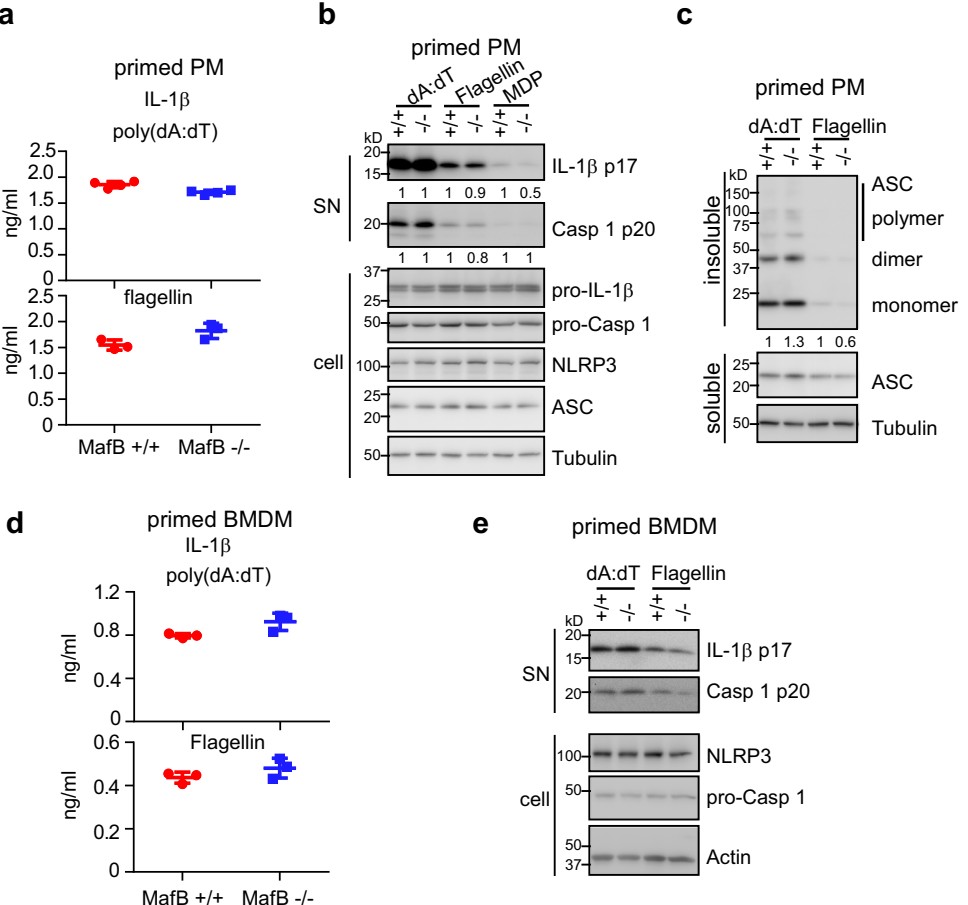

**Fig. 5 MafB does not participate in the NLRP1, NLRC4, or AIM2 inflammasome activation. a**, **b** Primed MafB+/+ and MafB−/− PMs were stimulated with the mix of LipoJet liposome and the AIM2 activator poly(dA:dT) (2 μg/ml), the NLRC4 activator Flagellin (1 μg/ml), or the NLRP1 activator muramyl dipeptide (MDP) (10 μg/ml). Levels of supernatant IL-1β were determined by ELISA. Levels of the indicated proteins in the supernatants and cells were determined by western blotting. **c** Primed MafB+/+ and MafB−/− PMs were treated as in (**a**, **b**). Insoluble extracts were cross-linked and ASC oligomerization determined. ASC in the soluble extracts was used as loading controls. **d**, **e** Primed MafB+/+ and MafB−/− BMDMs were stimulated with poly(dA:dT) or Flagellin. Supernatant IL-1β was determined by ELISA and the indicated proteins in the supernatants and cells were determined by western blotting. Mean ± SD. Representative of three independent experiments.

BMDMs with either MafB knockdown or knockout (Supplementary Fig. 3a, b, d). Together, these data suggest that p62 expression to the fullest level requires MafB in macrophages. Importantly, similar to MafB deficiency, p62 knockdown augmented the NLRP3 inflammasome activation, while had no effect on the LPS induced expression of pro-IL-1β and other pro-inflammatory cytokines in J774A.1 macrophages, BMDMs, and PMs (Fig. 7d–g and Supplementary Fig. 3c, e, f). Given the key role of p62 in restricting mitochondrial damage during the NLRP3 inflammasome activation, these data suggest that MafB regulates the NLRP3 inflammasome activation by sustaining p62 expression and the p62 protection of mitochondrial homeostasis.

**MafB upregulation diminishes the NLRP3 inflammasome activation**. Recently, retinoid X receptor (RXR) agonists have been reported to induce *Mafb* mRNA expression[29,37,38]. RXR is a member of the nuclear receptor transcription factor family. RXR has a wide variety of physiological functions, including cell differentiation, cell death, development, and lipid and glucose metabolism[44,45]. Activation of RXRs has been also shown to have an anti-inflammatory role, although the underlying mechanism is less defined[46–48]. As we have shown that MafB knockdown or knockout boosts the NLRP3 inflammasome activation, we were

intrigued to determine if upregulation of MafB yields an opposite effect. Macrophages were pre-treated with the RXR activator LG100268 (LG268)[29] and primed with LPS, followed by stimulation with ATP or Nigericin. We confirmed that LG268 does increase MafB at the protein level in macrophages (Fig. 8a), consistent with the fact that there is a RXR responsive element in the promoter of the *MafB* gene[37]. We found that LG268 has no effect on the LPS induced expression of pro-IL-1β and NLRP3 or pro-inflammatory cytokines TNF-α and IL-6 (Fig. 8a, b). However, LG268 markedly diminished the ATP and Nigericin induced mature IL-1β and Caspase 1 (Fig. 8a), suggesting that RXR upregulated MafB regulates the NLRP3 inflammasome. Furthermore, in accordance with the absent influence of the MafB knockdown and knockout, LG268 did not affect AIM2 or NLRC4 inflammasome activation either (Fig. 8a). To unequivocally demonstrate the effect of LG268 on NLRP3 inflammasome is dependent on MafB induction, we performed similar experiments in MafB−/− macrophages and found that the effect of this activator on the NLRP3 inflammasome activation is largely abolished (Fig. 8c). Taken together, these data suggest that RXR induced MafB specifically targets the NLRP3 inflammasome.

We showed that p62 expression requires MafB in macrophages. Given our finding that RXR activation induces MafB, we then determined the effect of LG268 on p62 expression. As shown in

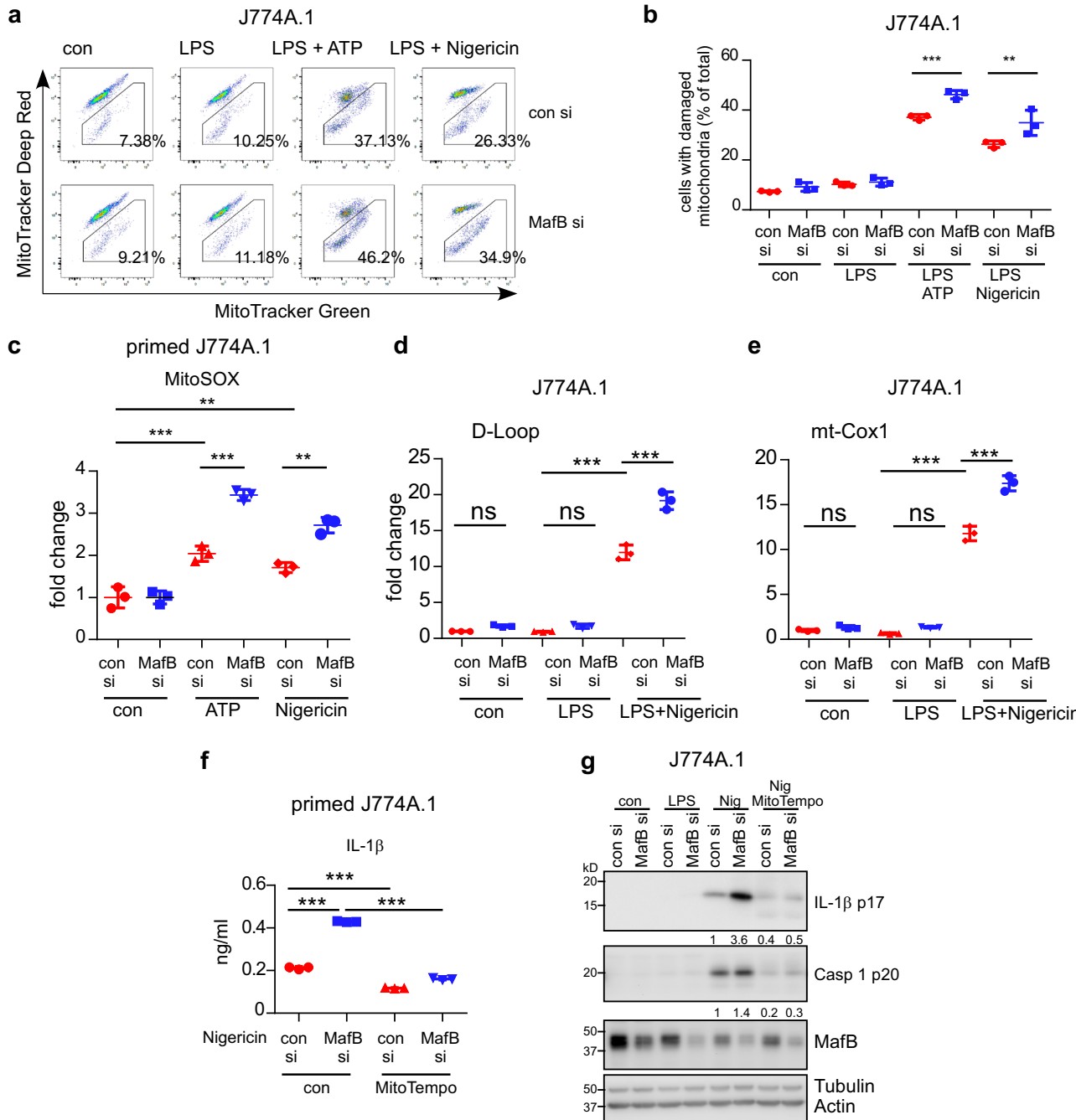

**Fig. 6 MafB restricts the NLRP3 agonists induced mitochondrial damage. a**, **b** J774A.1 macrophages were transfected with control siRNAs or MafB siRNAs. 48 h later, cells were treated with or without 100 ng/ml LPS for 4 h, ATP (5 mM) or Nigericin (10 μM) for 30 min, and stained with 100 nM MitoTracker Deep Red and 100 nM MitoTracker Green for additional 20 min. Cells were collected, and flow cytometric analysis was performed to determine the mitochondria damage. **c** Primed J774A.1 macrophages transfected with control siRNAs or MafB siRNAs were incubated with 2.5 μM MitoSOX for 15 min, followed by stimulation with ATP or Nigericin for 30–60 min. Mitochondrial ROS (mtROS) was determined by fluorescence microplate reader under 530/590 nm (Ex/Em). **d**, **e** J774A.1 macrophages transfected with control siRNAs or MafB siRNAs were primed with LPS for 4 h, followed by stimulation with Nigericin for 30 min. The cells were permeabilized with 25 μg/ml Digitonin for 10–15 min and cytoplasmic DNAs were prepared. Levels of mtDNA were determined by real-time PCR assay. **f**, **g** J774A.1 macrophages transfected with control siRNAs or MafB siRNAs were treated with LPS for 4 h, followed by MitoTempo (100 μg/ml) treatment for 30 min. The cells were stimulated with Nigericin for 60 min. Supernatant IL-1β was determined by ELISA (**f**), and levels of the indicated proteins in the supernatants and cells were determined by western blotting (**g**). Vertically stacked bands originate either from a single membrane or from a replicate membrane, with the same sample loadings. Mean ± SD; **$P < 0.01$, ***$P < 0.001$. Representative of three independent experiments. ns not significant.

Fig. 8d and in line with the MafB induction, LG268 effectively upregulated p62 in macrophages. The upregulation was dependent on MafB since the p62 induction by LG268 was abated in macrophages with MafB knockdown and knockout macrophages

(Fig. 8e, f). LG268 elevation of p62 was also concordant with the protective role of p62 in the NLRP3 activator induced mitochondrial damage, as evidenced by the ameliorated mitochondrial damage and ROS production in the LG268 treated macrophages

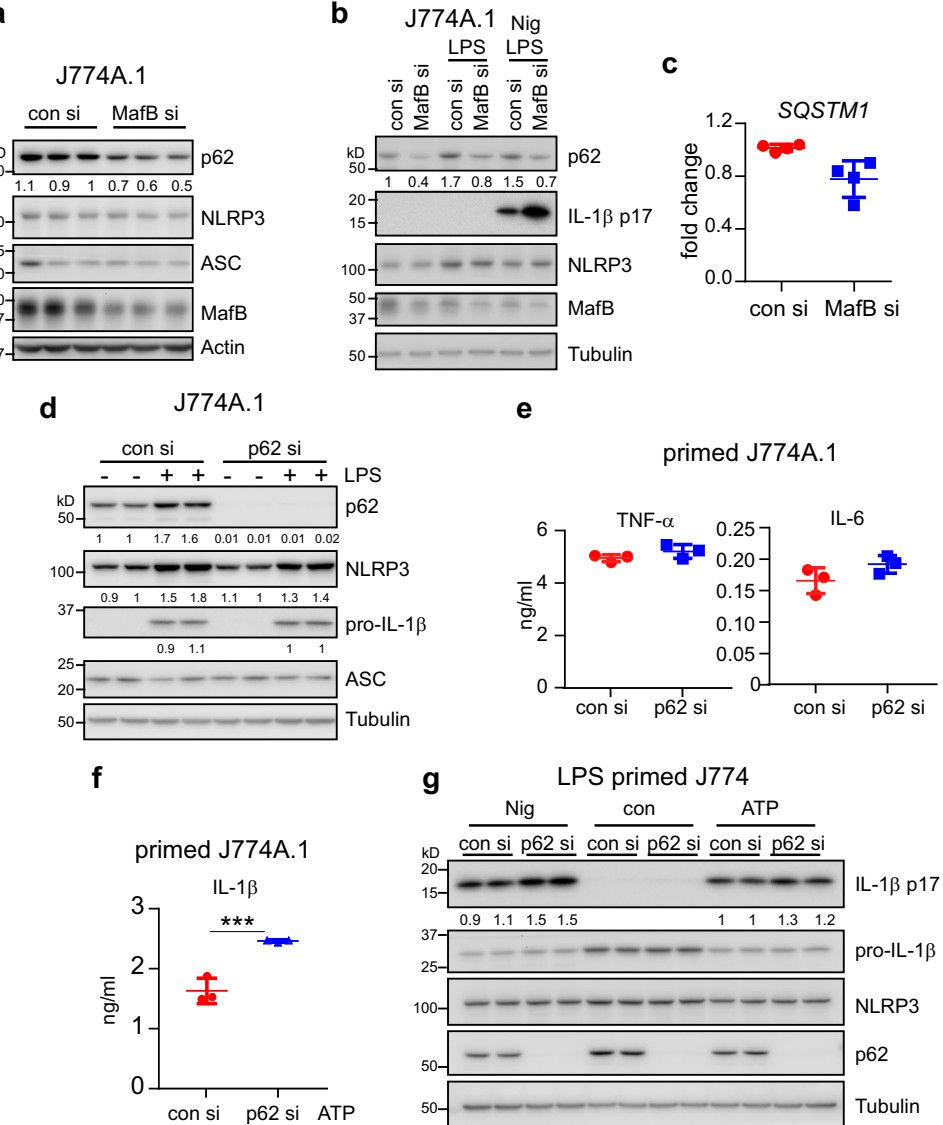

**Fig. 7 MafB sustains p62 expression in macrophages. a** J774A.1 macrophages were transfected with 50 nM control siRNAs or MafB siRNAs. 48 h after transfection, cells were collected, and protein levels of the indicated genes were determined by western blotting. **b** Levels of the indicated proteins in control siRNAs or MafB siRNAs transfected macrophages treated with LPS for 4 h, followed by Nigericin stimulation. **c** Experiment was done as in (**a**) and levels of *SQSTM1* (*p62*) transcription was determined by real-time PCR. **d, e** Levels of the indicated proteins in control siRNAs or p62 siRNAs transfected macrophages treated with or without LPS for 4 h (**d**) and the levels of supernatant TNF-α and IL-6 (**e**). **f, g** Levels of supernatant IL-1β (**f**) and the indicated proteins (**g**) in control siRNAs or p62 siRNAs transfected macrophages treated with LPS for 4 h, followed by Nigericin or ATP stimulation. Mean ± SD; ***P < 0.001. Representative of three independent experiments.

upon the NLRP3 inflammasome activation (Fig. 8g, h). Together, these data lend further support to the NLRP3 inflammasome regulating role of MafB. This finding also indicates that the anti-inflammatory role of RXR is, at least partially, mediated by MafB.

**Mye MafB−/− mice demonstrate elevated LPS and *P. aeruginosa* induced systemic and pulmonary IL-1β production and deficient lung *P. aeruginosa* clearance in vivo.** We have firmly established that MafB regulates the NLRP3 inflammasome activation in macrophages, we next asked if this gene has the same function in vivo. To test this hypothesis, we employed the mouse endotoxemia model by intraperitoneal injection of LPS, which has been well described leading to the NLRP3 inflammasome activation[20,49–52]. As shown in Fig. 9a–c, there was markedly increased IL-1β in the sera and lungs of Mye MafB−/− mice

with i.p. injection of LPS, which was concordant with enhanced activation of NLRP3 inflammasome in these lungs compared to MafB+/+ animals. In contrast, there was no significant difference in serum KC between these two strains. Since we found that HKPA rapidly downregulates MafB in macrophages. *P. aeruginosa* infection induced acute lung injury (ALI) has also been found to be dependent, at least partially, on NLRP3 activation[53]. We hypothesized that MafB regulates NLRP3 activation in *P. aeruginosa* pneumonia. As shown in Fig. 9d, there was significantly increased pulmonary IL-1β in Mye MafB−/− mice infected with live PAK. Upregulated KC levels could be due to the augmented inflammation resulted from the increased inflammasome activation. Protein level in the BAL from PAK infected Mye MafB−/− mice was also increased compared to the PAK infected MafB+/+ animals (Fig. 9e). Furthermore, PAK clearance in the lungs of Mye MafB−/− mice was impaired compared to that in the WT controls

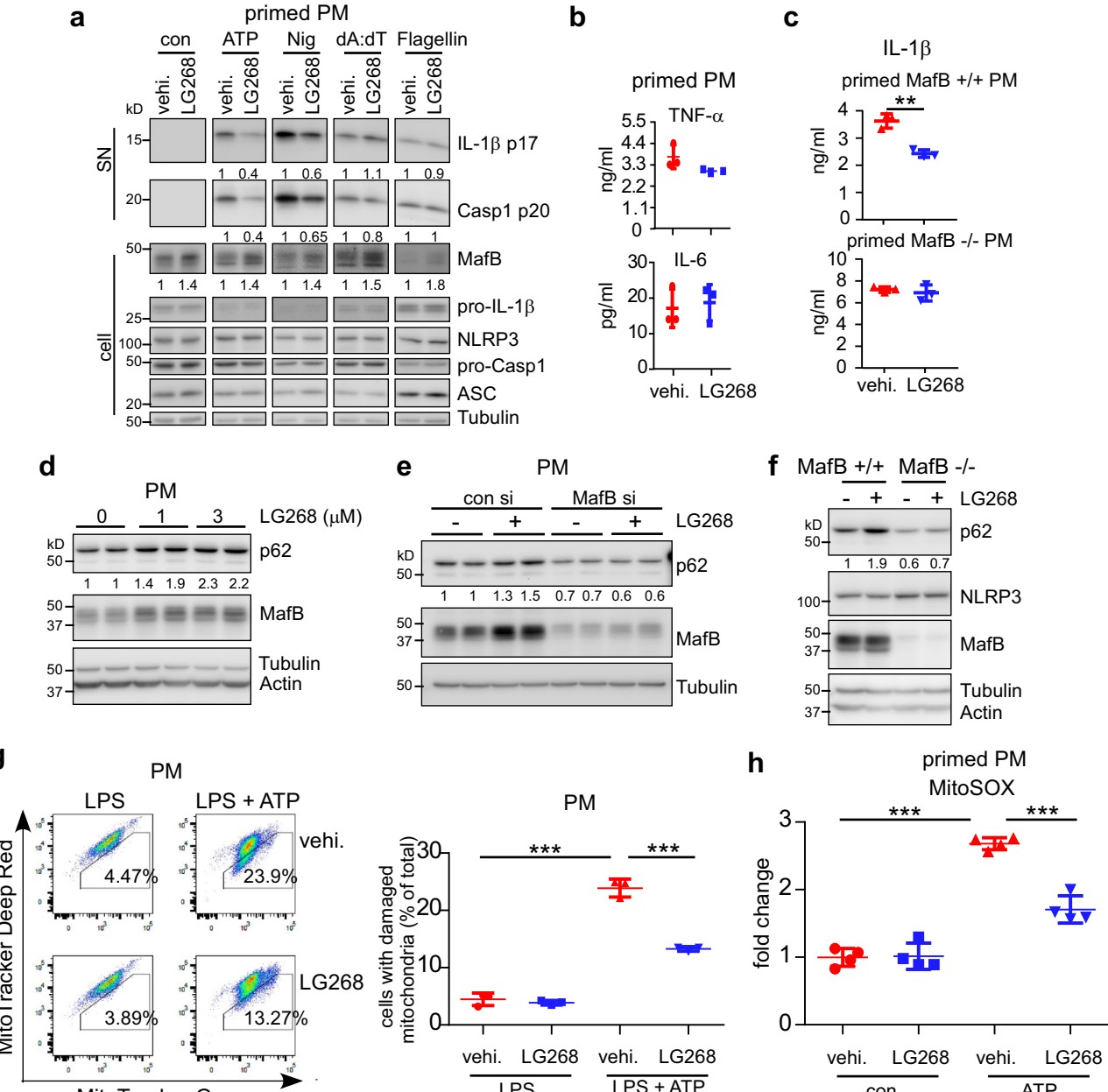

**Fig. 8 MafB upregulation diminishes the NLRP3 inflammasome activation. a, b** PMs were pre-treated with vehicle or the RXR activator LG100268 (LG268) (1 μM) for 1 day, followed by LPS treatment for 3 h. The cells were then treated with or without ATP (5 mM) or Nigericin (5 μM), or transfected with poly(dA:dT) or Flagellin. Levels of the indicated protein in supernatants and cell extracts was determined by western blotting (**a**). Levels of supernatant TNF-α and IL-6 were determined by ELISA (**b**). Vertically stacked bands originate either from a single membrane or from a replicate membrane, with the same sample loadings. **c** MafB+/+ and MafB−/− PMs were pre-treated with LG268, followed by LPS priming and ATP activation. ELISA for IL-1β was performed. **d** MafB +/+ PMs were treated with various concentrations of LG268 for 1 day. Western blotting was performed for the indicated proteins. **e** Levels of the indicated proteins in control siRNAs or MafB siRNAs transfected PMs treated with or without LG268 for 1 day. **f** MafB+/+ and MafB−/− PMs were treated with or without LG268 for 1 day. **g** PMs were pre-treated with vehicle or the RXR activator LG268 for 1 day, followed by LPS treatment for 4 h. The cells were then stimulated with ATP for 30 min, and stained with 100 nM MitoTracker Deep Red and 100 nM MitoTracker Green for additional 20 min. Cells were collected, and flow cytometric analysis was performed to determine the mitochondria damage. **h** Primed PMs were incubated with MitoSOX for 15 min, followed by ATP stimulation. mtROS levels were determined. Mean ± SD; **P < 0.01, ***P < 0.001. Representative of three independent experiments.

(Fig. 9f). Given macrophages from MafB+/+ and Mye MafB−/− mice displayed similar PAK killing activities in vitro (Fig. 9g), the impaired PAK clearance in the lungs of Mye MafB−/− mice was likely due to the enhanced NLRP3 inflammasome activation and the worsened lung injury in these mice. Together, our data have established that MafB regulates NLRP3 inflammasome both in vitro and in vivo, which leads to mitigated NLRP3 associated ALI.

## Discussion

The NLRP3 inflammasome is unique in that its activation is an orderly two-step process, e.g., priming and activating[1,3,4,13,14]. There are several events that have been identified in the priming stage are critical to its subsequently full activation, including the upregulated expression of NLPR3 and pro-IL-1β, and certain post-transcription modifications on NLRP3[1,3,4,13,14]. There are also a

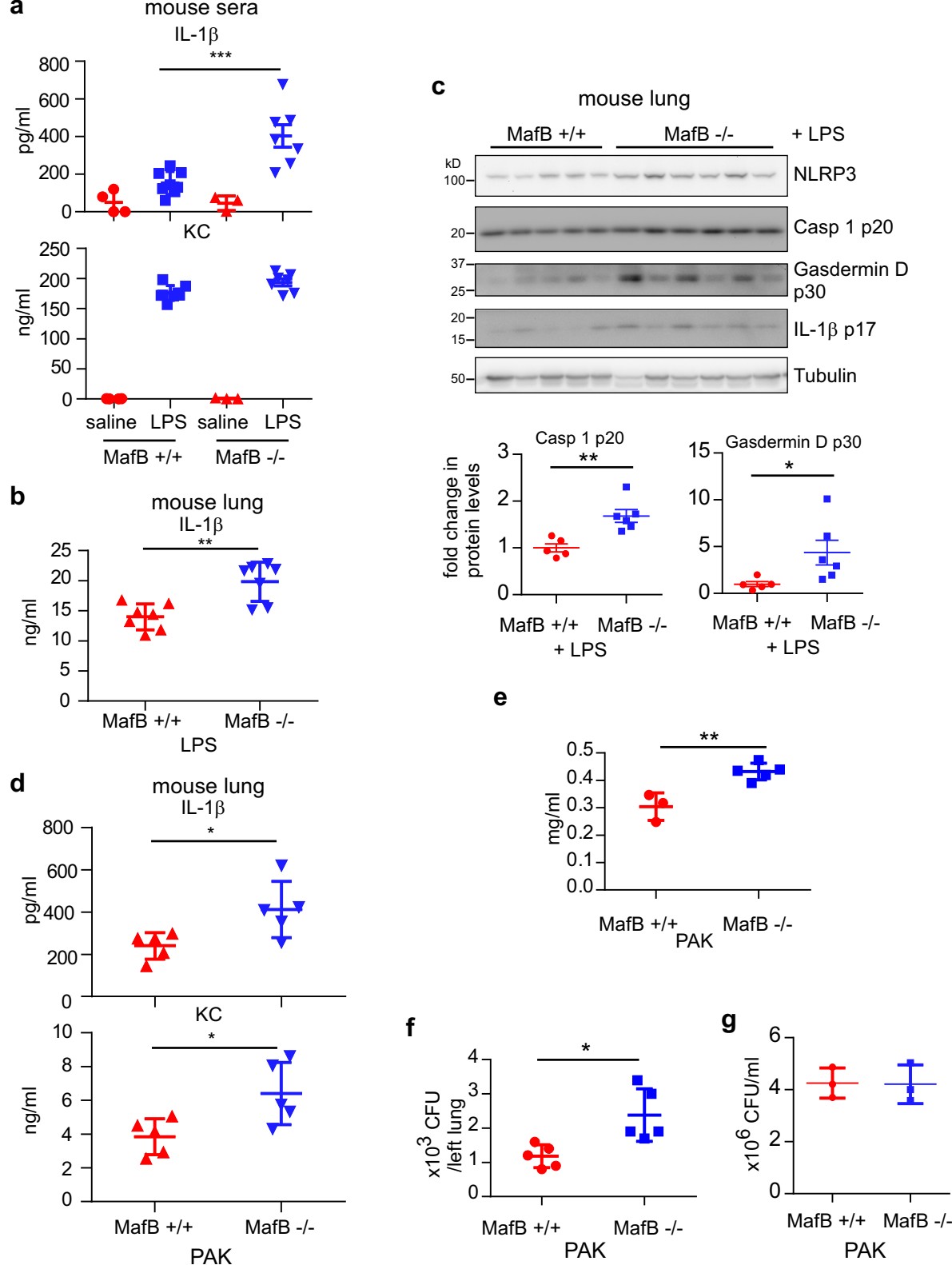

**Fig. 9 Mye MafB −/− mice demonstrate elevated LPS and *P. aeruginosa* induced systemic and pulmonary IL-1β production and deficient lung *P. aeruginosa* clearance in vivo. a–c** Ten-week MafB fl/fl and Mye MafB−/− mice were injected i.p. saline or LPS (10 mg/kg). Three hours after injection, mice were sacrificed and sera and lung extracts prepared. Levels of the indicated cytokines were determined by ELISA (**a**, **b**). Levels of the indicated protein in the lung extracts were determined by western blotting (**c**). **d–f** Ten-week MafB fl/fl and Mye MafB−/− mice were i.t. instilled with 0.6 × 10⁷ PAK. 24 h after infection, lung extracts were prepared and levels of the indicated cytokines were determined by ELISA (**d**); BAL protein levels were measured (**e**); and lung extracts were diluted and plated. Bacterial colonies were counted to determine CFU (**f**). **g** MafB+/+ and MafB−/− BMDMs in 96-well plates were incubated with 2 × 10⁶ PAK for 1 h. The cells were lysed in 100 µl sterile H2O for 20 min and combined with supernatants for serial dilution and plating. Bacterial colonies were counted to determine CFU. *n* = 3–8; mean ± SD or SE; *P < 0.05, **P < 0.01, ***P < 0.001. PAK *P. aeruginosa* strain K, CFU colony-forming unit.

number of proteins that have been found to control the subsequent activation via regulating the assembly of the inflammasome machinery or other mechanisms[1,3,4,13,14]. In this study, we found that MafB transcription is abruptly downregulated and the MafB protein is quickly degraded by proteasome in the priming stage. We also demonstrated that MafB is a negative regulator of the NLRP3 inflammasome activation via sustaining the expression of p62 and restricting mitochondrial damage inflicted by the NLRP3 agonists. Therefore, our study indicates that MafB downregulation in the priming phase is a fine-tuning step for the maximum activation of the NLRP3 inflammasome.

Despite we showed that the MafB protein level is critically controlled by the ubiquitin-proteasome system, it would be very significant to delineate the underlying mechanism associated with this process. It would be also therapeutically essential to identifying the MafB ubiquitinase(s). First, understanding these processes sheds a new light into how the ubiquitin system regulates the NLRP3 inflammasome, as it has been shown to be the integral part of the NLRP3 activation[54,55]. Second, it opens new avenues of investigation that harness the anti-NLRP3 inflammasome activity of MafB by small molecule compounds targeting the ubiquitin-proteasome-mediated degradation of this protein to treat pathologies where the hyperactivation of this type of inflammasome is prevalent. Together, agents that transcriptionally upregulate MafB, such as LG268 and MafB stabilizing compounds may be proven an effective class of novel therapeutics for inflammatory disorders.

Although we have established that MafB is required for p62 expression to the fullest extent in various conditions, the underlying mechanism remains unclear and warrants further investigation. Our data that both p62 expression at the basal line and p62 induction by LPS require MafB do not support a role of MafB in regulating the LPS activated transcriptional events, which is consistent with the finding that MafB did not participate in the induction of NLRP3 and pro-IL-1β in the priming stage. Additionally, we did not find that MafB considerably regulates the transcription of p62 itself, indicating that this regulation involves a direct or indirect mechanism at the posttranscriptional and the translational levels. Nevertheless, a thorough investigation of MafB regulation of *p62* would be very significant because not only it will delineate how MafB specifically regulates the NLRP3 inflammasome activation, but also it will shed a light on new mechanisms underlying the deregulation of autophagy/mitophagy frequently found in many pathological conditions.

We have shown that the regulation of the NLRP3 inflammasome by MafB is associated with p62 expression and likely its mediated mitophagic clearance of the damaged mitochondria induced by NLRP3 agonists. It remains possible that alternative mechanisms are involved. Indeed, NLRP3 undergoes multiple posttranslational modifications in the priming stage, such as ubiquitination and phosphorylation, which have been shown to be critical regulatory mechanisms of the NLRP3 inflammasome activation[54,55]. Since our findings show that MafB degradation occurs at the priming step and the turnover of this protein is entirely mediated by ubiquitin-proteosome machinery, a potential regulation of the NLRP3 posttranslational modifications by MafB and/or MafB-associated proteins is both plausible and intriguing. Although this hypothesis was not the focus of this study, a further investigation into this scenario will definitely improve the understanding of the complexity involved in the NLRP3 inflammasome priming and activation sequential events.

The RXR signaling is known to regulate inflammatory response, which is generally mediated by transcriptional modifications of the pro-inflammatory targets by RXR and associated other transcriptional partners, such as PPAR[46–48]. Additionally, the activation of RXR/LXR/PPAR promotes efflux of intracellular lipids, some of which are quite pro-inflammatory. Intracellular lipids can be also degraded by lipases and lipophagy in certain conditions[56,57]. In this study, we surprisingly found that the RXR activator LG268 increases p62 expression, which is dependent on the upregulation of MafB in macrophages. p62, as it is required by mitophagy, has been shown to also participate in lipophagy[56,57]. Therefore, our findings here suggest multiple functional and mechanistic significances in MafB, including a novel function of MafB in mediating lipophagy, and another plausible mechanism for the MafB regulation of the NLRP3 inflammasome activation due to it enhancing lipophagy by promoting p62 expression. These hypotheses will remain front and center focus of our future studies.

In conclusion, we showed that a rapid proteasome-mediated degradation of MafB in the priming stage constitutes a fine-tuning step leading to the NLRP3 inflammasome activation in macrophages. We presented evidence that MafB sustains p62 expression, and restricts the NLRP3 agonists triggered mitochondrial damage and the resulted mitochondrial ROS production and DNA release, thereby MafB specifically regulating the NLRP3 inflammasome. Targeting MafB, either promoting its stability and/or increasing its transcription through small molecule compounds, has the potential to be novel therapeutics to treat many inflammatory disorders caused by excessive activation of the NLRP3 inflammasome.

Finally, we acknowledge that there are notable limitations in the study. First, we found that p62 expression was reduced in MafB knockdown and knockout macrophages treated with or without LPS. However, LPS also led to rapid p62 upregulation. The current data have not provided insight into the biological implication of these two antagonistic events in the NLRP3 activation, partly because the study did not establish the dynamic of MafB regulating p62 expression. These dueling actions may be one of the ways for the macrophages to fine tune the activation of NLRP3 inflammasome. Second, although our findings indicate that p62 may mediate MafB regulation of the NLRP3 activation, the causality between these two events remains to be fully established. Additionally, the effect of p62 was generally milder than MafB itself in this process, suggesting the involvement of other mitophagic mediators.

## Methods

**Reagents**. Ultra-pure LPS from *Escherichia coli* O111:B4 was from Sigma-Aldrich. LG100268 and MLN7243 was from Cayman. Nigericin, muramyl dipeptide (MDP), Flagellin, poly(dA:dT) and heat-killed *P. aeruginosa* (HKPA) were from Invivogen. ATP, cycloheximide (CHX), MG132, and Bafilomycin A1 were from Sigma-Aldrich. CytoTox 96® Non-Radioactive Cytotoxicity LDH Assay was from Promega. LipoJet™ In Vitro Transfection Kit was from SignaGen Laboratories.

**Mice**. C57BL/6 mice and myeloid lineage Cre mice Lyz2 Cre/Cre were purchased from The Jackson Laboratory. MafB fl/fl mice were gifts from Dr. Lisa Goodrich at Harvard University[26]. Mice with ablation of myeloid MafB, MafB mye−/−, were established by cross-breeding MafB fl/fl with the Lyz2 Cre/Cre lines. In all, 8–10 weeks male mice were used for all experiments.

**Cell lines**. Mouse macrophage cell line J771A.1 was from American Type Culture Collection (Manassas, VA).

**Mouse bone marrow-derived macrophages (BMDMs) and peritoneal macrophages (PMs)**. Mouse BMDMs were established from bone marrow cells of C57BL/6 mice as previously described[58]. Briefly, bone marrow cells were cultured in DMEM containing 10% FBS and 50 ng/ml murine macrophage colony-stimulating factor (M-CSF) (R&D Systems) for 5 days. The differentiated cells were

then split and plated for following experiments. To prepare peritoneal macrophages, 8–10-week mice were intraperitoneally (i.p.) injected with 1 ml 4% Brewer thioglycollate (Sigma-Aldrich). After 4 days of elicitation, peritoneal cells were collected by lavage with cold DMEM and plated for 2 h, followed by extensive wash to remove non-adherent cells. The adherent cells were used as peritoneal macrophages.

**LPS-induced endotoxemia**. Mice were injected i.p. LPS (10 mg/kg body weight in 50 µl saline). Three hours after i.p., mice were sacrificed. Mouse whole blood was collected via abdominal aorta and sera prepared. Lung extracts were prepared.

**P. aeruginosa lung infection**. Freshly growing *P. aeruginosa* strain K (PAK) in exponential phase were washed, counted and re-suspended in PBS. Mice were i.t. instilled with $0.6 \times 10^7$ live PAK in 50 µl saline. Twenty-four hours after infection, mice were sacrificed and bronchoalveolar lavage (BAL) was performed. Left lungs were homogenized in 1 ml PBS. 100 µl of the 1:10 dilution of the lung homogenates were plated on LB-Agar and cultured at 37 °C overnight. Bacterial colonies were counted by ImageJ program and colony-forming unit (CFU) was calculated.

**Bacteria killing assay**. Freshly growing PAK in exponential phase were washed, counted and re-suspended in antibiotics-free DMEM containing 10% FBS. In total $2 \times 10^6$ PAK was added to macrophages cultured in 96-well plates and incubated at 37 °C for 1 h. After supernatants collection, the cells were lysed in 100 ul sterile H2O for 20 min and then combined with the supernatants for serial dilution and plating on LB-agar. The plates were incubated at 37 °C overnight, and bacterial colonies were counted by ImageJ (NIH). Data were presented as CFU/ml.

**siRNA transfection**. siRNA transfection was performed using HiPerFect reagents (Qiagen) according to the manufacturer's instructions. ON-TARGETplus negative control siRNA pool, specific mouse MafB siRNA pool and p62 siRNA pool were from Dharmacon.

**RNA isolation and real-time PCR**. Total RNA was extracted using Qiagen RNeasy Mini Kit. After reverse transcription, mRNA levels were determined by real-time PCR using SsoAdvanced™ Universal SYBR® Green Supermix (Bio-Rad) on Bio-Rad CFX384 Real-Time PCR System. Primer sequences for mouse genes are: Tubulin α1: sense, 5'-GGATGCTGCCAATAACTATGCTCGT-3', antisense, 5'-GCCAAAGCTGTGGAAAACCAAGAAG-3'; MafB: sense, 5'-CACACACACGAAGAAAACAAGACAG-3', antisense, 5'-CAGATCATCACACGTAGCAAGAGGT-3'; IL1b: sense, 5'-AAGGAGAACCAAGCAACGACAAAATA-3', antisense, 5'-TTTCCATCTTCTTCTTTGGGTATTGC-3'; TNFA: sense, 5'-AAATTCGAGTGACAAGCCTGTAGCC-3', antisense, 5'-GTTGGTTGTCTTTGAGATCCATGCC-3'; IL6: sense, 5'-CCCAATTTCCAATGCTCTCCTA-3', antisense, 5'-AGGAATGTCCACAAACTGATATGCT-3'; NLRP3: sense, 5'-GAGACCGTGAGGAAAGGACCAGGA-3', antisense, 5'-GCCAAAGAGGAATCGGACAACAAA-3'; SQSTM1/p62: sense, 5'-ACACGATCCAGTATTCGAAGCACC-3', antisense, 5'-AAAAGGCATCACACATTCCCACCT-3'. To calculate fold change in the expression of these genes, $\Delta Ct = Ct$ of Tubulin - Ct of individual genes was first obtained. $\Delta\Delta Ct = \Delta Ct$ of treated groups - $\Delta Ct$ of untreated control groups was then obtained. Fold change was calculated as $2^{\Delta\Delta Ct}$, with control groups as 1.

**Western blotting**. Western blotting was performed as previously described[59]. Briefly, proteins were separated by 6–15% SDS-PAGE

and electro-transferred to PVDF membranes. The membranes were blocked in TBST with 5% BSA for 30 min at room temperature, incubated with primary antibodies overnight at 4 °C, followed by incubation with HRP-conjugated secondary antibodies (Bio-Rad) for 1 h at room temperature. After washing, blots were developed using the SuperSignal™ West Dura Extended Duration Substrate (Thermo Scientific) and imaged on a Bio-Rad ChemiDoc Imaging system. Mouse anti-α-tubulin antibody (T5168) and mouse anti-β-actin antibody (A1978) were from Sigma-Aldrich. Rabbit anti-ubiquitin (#3936), anti-cleaved-IL-1β (#63124), anti-NLRP3 (#15101), anti-ASC (#67824), anti-Gasdermin D (#93709) and anti-p62 (#39749) antibodies were from Cell Signaling Technology. Mouse anti-NLRP3 (AG-20B-0014-C100) and anti-Caspase 1 (p20) (AG-20B-0042-C100) antibodies were from Adipogen. Goat anti-IL-1β antibody (AF-401-NA) was from R&D Systems. Rabbit anti-MafB (20189-1-AP) and mouse anti-GAPDH (60004-1-Ig) were from Proteintech. All antibodies were used at a dilution of 1:2000. Protein densitometric analyses were performed using ImageJ using α-tubulin or β-actin as reference.

**Immunoprecipitation (IP)**. IP was performed using Dynabeads™ Protein G (Invitrogen) according to the manufacturer's instructions. Briefly, 2 µg rabbit anti-MafB antibody diluted in 200 µl PBS was incubated with 10 µl Dynabeads Protein G for 10 min at room temperature. Macrophages were lysed in IP lysis buffer (1% NP-40, 1 mM EDTA, 25 mM Tris-HCl (pH 7.4), 150 mM NaCl and 5% glycerol) supplemented with protease and phosphatase inhibitor cocktails (Sigma-Aldrich), followed by centrifugation at $12,000 \times g$ for 10 min at 4 °C. The cellular lysates were then incubated with the antibody bound Dynabeads overnight at 4 °C. After incubation, beads were washed three times with IP lysis buffer. Proteins were eluted from beads with 2×SDS sample buffer, boiled, and separated by SDS-PAGE for western blotting.

**Immunofluorescence staining**. Macrophages were seeded and cultured on glass coverslips in 24-well plates. After treatment, the cells were washed twice with PBS, fixed with 4% paraformaldehyde (PFA) for 15 min, permeabilized with 0.1% Triton X-100 in PBS for 5 min, and blocked in PBS containing 1% BSA for 1 h. The cells were then incubated overnight with primary antibodies (1:100 dilution) at 4 °C, followed by incubation with fluorochrome-conjugated secondary antibodies (1:500 dilution, ThermoFisher) for 1 h at room temperature. After wash, nuclei were counterstained with DAPI containing mounting medium and the cells were imaged using the Olympus IX73 Microscope System.

**Enzyme-linked immunosorbent assay (ELISA)**. Levels of IL-1β, IL-18, TNF-α, IL-6 and keratinocyte-derived chemokine (KC) in cell culture supernatants, mouse sera, or lung tissue extracts were determined using DuoSet ELISA development kits (R&D Systems) according to the manufacturer's instructions.

**Determination of ASC oligomerization**. Treated macrophages were collected in cold 0.5% Triton X-100 PBS. The cell suspension was vortexed once and kept on ice for 10 min, followed by centrifugation at $6840 \times g$ for 15 min. The Supernatants were transferred to new tubes and used as soluble fractions. The pellets were washed twice using cold PBS and cross-linked with 2 mM BS3 (bis(sulfosuccinimidyl)suberate) (ThermoFisher) at room temperature for 30 min. The cross-linked pellets were again centrifuged at $6797 \times g$ for 10 min and lysed directly in 2×SDS sample buffer.

**Determination of mitochondrial reactive oxygen species (mtROS)**. Mitochondrial ROS were determined using MitoSOX Red (Invitrogen) according to the manufacturer's instructions. Briefly, macrophages were first primed with LPS followed by incubation with 2.5 μM MitoSOX Red for 20 min at 37 °C. The cells were then stimulated with ATP or Nigericin for 30–60 min, and fluorescence intensity was determined using a multi-mode microplate reader (BioTek).

**Determination of mitochondrial damage**. Mitochondrial damage was evaluated by a flow cytometry-based assay as previously described[24]. In brief, macrophages were first primed with LPS, followed by ATP or Nigericin treatment for 30 min, after which the cells were stained with 100 nM MitoTracker Deep Red and 100 nM MitoTracker Green (Cell Signaling Technology) for 15–30 min. The cells were then washed and analyzed on a BD FACSymphony A3 Cell Analyzer. Data analysis was performed using FlowJo version 10 software (Tree Star). Gating strategy for flow cytometric analysis is shown in Supplementary Fig. 4.

**Mitochondrial DNA (mtDNA) quantification**. Macrophages were permeabilized with 25 μg/ml digitonin for 15 min. Supernatants were collected and centrifuged at 500 g for 10 min at 4 °C. The supernatants were transferred to new tubes and centrifuged twice at 12,000×g for 10 min at 4 °C to remove nuclear and mitochondrial contamination from the cytosolic fraction. DNA was isolated from the cytosolic fractions using PureLink™ Genomic DNA Mini Kit (Invitrogen), and cytosolic mtDNA levels were determined by real-time PCR. Primer sequences are: mt-Cox1: sense, 5'-AATAATTGGAGGCTTTGGAAACTGA-3', antisense, 5'-GGAGAAGGAGAAATGATGGTGGTAG-3'; D-Loop: sense, 5'-AATCTACCATCCTCCGTGAAACC-3', antisense, 5'-TCAGT TTAGCTACCCCCAAGTTTAA-3'.

**Statistics and reproducibility**. Statistical analyses were performed using GraphPad Prism 8. One-way ANOVA followed by the Bonferroni test was used for multiple group comparisons. The two-tailed Student's $t$ test was used for comparison between two groups. $P < 0.05$ was considered statistically significant. $n = 3–4$ biological replicates for all experiments unless indicated otherwise in the Figure Legends.

**Study approval**. Protocols for all experiments involving mice in this study were approved by the University of Alabama at Birmingham Institutional Animal Care and Use Committee (IACUC). We have complied with all relevant ethical regulations for animal testing.

**Reporting summary**. Further information on research design is available in the Nature Portfolio Reporting Summary linked to this article.

## Data availability
Numerical source data underlying all graphs are available in the Supplementary Data 1. The uncropped/unedited western blot images are included in Supplementary Fig. 5. All other data are available from the corresponding author upon reasonable request.

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

## Acknowledgements

This work was supported by NIH grants R35HL135830 and R01AI170913 to G.L. and United States Department of Defense grant W81XWH-20-1-0226 to G.L.

## Author contributions

G.L. designed the study; H.C., S.B., N.X., and T.D. performed the cell and animal experiments; H.C., R.M.L., Y.Y.S., and G.L. analyzed the data and wrote the manuscript.

## Competing interests

The authors declare no competing interests.
