## [Peer Review File · Communications Biology]

Reviewers' comments:

Reviewer #1 (Remarks to the Author):

This manuscript identifies a role of MafB as a negative regulator of NLRP3 inflammasome responses, proposed to work through a mechanism consisting of stabilisation of p62 which prevented mitochondrial destabilisation. This is a novel pathway of NLRP3 regulation that has not been described previously, and as such will be of interest to the field. Whilst the data in this paper shows a clear effect of MafB in regulating NLRP3 inflammasome responses, there are a couple of points that if addressed would improve the manuscript to be suitable for publication in Communications Biology.

Major comments:

- 1) Densitometry of western blots would help strengthen the authors claims. Whilst some changes are clear, others are less convincing, such as the changes in p62. Densitometry should be performed on figures 1A-C (MafB, IL-1b, NLRP3), and for MafB and p62 throughout. This is important as it will demonstrate the change in these proteins, and show the reproducibility between several independent experimental repeats.
- 2) If LPS massively down-regulates MafB, it seems counter-intuitive that further knockdown would enhance NLRP3 (as there is not much to knockdown). Therefore, does NLRP3 activation prevent degradation of MafB, and so levels are restored following the activating stimulus, which is prevented by siRNA/KO? The authors should examine MafB and p62 levels following NLRP3 activation.
- 3) Evidence of MafB knockdown is missing in figure 3. It is important to show a representative level of knockdown in these experiments.
- 4) In Figure 6A, it is not explained why a large number of cells with mitotracker signal have been excluded, as they appear to be positive for both colours but ignored from the analysis.

Minor comments:

- 1) In the introduction: 'with (as for NLRP3 and AIM2) or without (as for NLRP1 and NLRC4) an assistance from the adapter ASC'. NLRP1 and NLRC4 can use ASC if present so this statement is not completely true.
- 2) In the methods, mtDNA quantification: methodology on subcellular fractionation is missing.
- 3) Quantification of ASC specks in 3E would be useful to show how consistent the response is between experimental repeats.
- 4) In the discussion: 'Additionally, we did not find that MafB considerably regulates the transcription of p62 itself (data not shown)'. This data should be included as it provides more mechanistic insight into the authors proposed mechanism.

Reviewer #2 (Remarks to the Author):

The manuscript "MafB inhibits the NLRP3 inflammasome activation by sustaining p62 expression in macrophages" describes a role for the transcription factor MafB as a negative regulator of the NLRP3 inflammasome. The mechanism proposed for this effect involves the expression of p62, which in turn promotes clearance of damaged mitochondria. While the manuscript is of good quality and has good amount of evidence supporting their claims, some points require further elaboration.

Major points:

- There are no details regarding immunoprecipitation. Please add in the methods how the immunoprecipitation was performed (lysis buffer and conditions, beads used, antibody dilutions, incubation times, elution method, etc.)

- Figure 5: Activation of NLRC4 and AIM2 is usually made by transfection of flagellin and poly(dA:dT) respectively. Were these reagents transfected into the cells? If so, how? Additionally, MDP is not a bona fide NLRP1 agonist (Chien-Hsiung Yu et. al., Journal of Molecular Biology, 2018). As it is currently presented, possible effects of MafB on NLRC4, AIM2 and NLRP1 cannot be discarded.
- Figure 8: While the mitoSOX data on figure 8G is very nice, it is not a direct indicator of mitochondrial damage, as changes in metabolism also affect this assay. The mitotracker Red/Green assay employed in Figure 6A would be better suited to study mitochondrial damage.
- Claims regarding MafB and p62 need to be softened. For example, the manuscript claims on page 17 "(...)we have firmly established that MafB is required for p62 expression(...)". However, figure 8 data suggests that MafB enhances p62 expression, but it is not required as cells knockdown or knockout for MafB express detectable amounts of p62.
- Figure 9 C,D,E: One alternative explanation for these data is that MafB^{-/-} animals have enhanced IL-1b due to increased bacterial burden. Additional data demonstrating that MafB does not impact bacterial clearance in vitro would make this alternative explanation less likely and strengthen the conclusions in this figure.

Minor points:

- Some of the conclusions rely on comparing expression levels using immunoblotting data. While in many cases these differences are visually striking, in some cases the visual differences are too small to draw conclusions. The addition of densitometric data would make these conclusions less ambiguous.
- Some of the text regarding Figure 4 is confusing. On page 11 "(...)the increased IL-1b in the supernatants was due the enhanced inflammasome activation, but not a heightened secretion, because there was more mature IL-1b in the combined cellular and supernatant extracts from Mafb^{-/-} macrophages (Figure 4E)." The presence of more mature IL-1b is usually associated with inflammasome activity, but figures 4B and 4D unambiguously shows that this IL-1b is secreted and found in the supernatants.
- Some graphs would benefit from more information on the y-axis. For example, most of Figure 9 would benefit from including what is being quantified in the y-axis (IL-1b pg/mL, etc.).
- Some acronyms are not defined in the manuscript (MAF, KC, ALI) while others are defined multiple times (PAK).
- The English in the manuscript needs some refinement.
- Information regarding LPS concentrations is missing in some figure legends.
- In the methods section, please add antibody dilutions for the various antibodies used for immunoblots.
- Methods: "Determination of ASC homorization". Is "homorization" correct, or should it be oligomerization? I could not find a definition for the term homorization.

Reviewer #3 (Remarks to the Author):

This article by Cui and colleagues interesting looks to the role of MafB in regulating NLRP3 inflammasome activity. They show, largely using cell culture models, that in the absence of MafB signaling, NLRP3 inflammasome activity is increased demonstrated by enhanced release of active IL-1beta and production of ASC specks.

I found the manuscript to be interesting and relevant for the field, although the following experiments are important to confirm the findings.

1. None of the western blots are quantified, and the blots should show both the full and cleaved bands for Caspase-1, IL-1b and Gasdermin D
2. Some western blots do not show the groups together on the same gel e.g. Fig 1F, Fig3D, Fig 8A

3. A number of experiments are missing the control, untreated groups, e.g. Fig 2B, 2D, 3A-C, 3G, 4A-D, 4I-J, Fig8A
4. Fig 3E is not quantified
5. Labelling is inconsistent e.g. sometimes the graph states LPS+ATP other times just ATP (e.g. Fig 6C-E). The graph sizes also vary within the same figures
6. For Fig 7, the authors state that "p62 knockdown diminished the NLRP3 inflammasome activation..." (page 14) however in Fig 7E, the levels of IL-1b are actually significantly increased after p62 siRNA treatment, and the bands on the westerns on 7F also looks stronger. Can the authors comment on this?
7. If ubiquitination is as necessary as the authors suggest, an experiment inhibiting ubiquitination would be helpful for the manuscript. The westerns blots provided in Fig 1K are hard to interpret
8. It is surprising the authors did not make use of western blots for their in vivo experiment in Fig 9. I suggest to add those as without confirming the NLRP3 inflammasome components, the IL-1beta could be coming from other pathways
9. There are some typos throughout e.g. p12 "Gasdermine" and "oilgomerization" I suggest going through the manuscript in detail with a spell checker
10. There is a recent paper from Billingham et al 2022 that shows mitochondrial ROS is not necessary for NLRP3 activation. This finding, and others showing mitochondria-independent mechanisms for NLRP3 inflammasome activation should be detailed more in the introduction, which currently reads that basically all NLRP3 inflammasome triggers work by inducing mitochondrial disruption.
11. In the discussion the authors write "MafB is an important factor in both the priming and activating phases during the NLRP3 inflammasome activation". They do show that MafB has a role in the NLRP3 assembly (i.e. increased IL-1beta and production of ASC specks), but how it may affect NLRP3 priming is not clear. Indeed they also comment that "MafB specifically inhibiting the NLRP3 inflammasome". This seems not accurate, as if it was true inhibition, MafB overexpression would block NLRP3 activation. Rather, MafB seems to be regulating, or fine tuning NLRP3 inflammasome activity, and I suggest updating the text to reflect that

Reviewer #1:

General comments:

This manuscript identifies a role of MafB as a negative regulator of NLRP3 inflammasome responses, proposed to work through a mechanism consisting of stabilization of p62 which prevented mitochondrial destabilisation. This is a novel pathway of NLRP3 regulation that has not been described previously, and as such will be of interest to the field. Whilst the data in this paper shows a clear effect of MafB in regulating NLRP3 inflammasome responses, there are a couple of points that if addressed would improve the manuscript to be suitable for publication in *Communications Biology*.

Response: We appreciate the reviewer's encouraging comments and insightful suggestions. We have tried our best to revise the manuscript as suggested by the reviewer.

Comment 1: Densitometry of western blots would help strengthen the authors claims. Whilst some changes are clear, others are less convincing, such as the changes in p62. Densitometry should be performed on figures 1A-C (MafB, IL-1b, NLRP3), and for MafB and p62 throughout. This is important as it will demonstrate the change in these proteins, and show the reproducibility between several independent experimental repeats.

Response 1: We have performed the densitometry analyses on all key Western blotting images that are essential to the central conclusions. We would like to share that all experiments in the manuscript were done at least two times and representative results are shown.

Comment 2: If LPS massively down-regulates MafB, it seems counter-intuitive that further knockdown would enhance NLRP3 (as there is not much to knockdown). Therefore, does NLRP3 activation prevent degradation of MafB, and so levels are restored following the activating stimulus, which is prevented by siRNA/KO? The authors should examine MafB and p62 levels following NLRP3 activation.

Response 2: We appreciate the reviewer's insight. We examined the levels of MafB and p62 following the treatments with the NLRP3 agonists. However, we found that the NLRP3 activation did not prevent or reverse the MafB degradation or p62 expression (Figures 3D, 3H, 4E, 6G, 7B, 7G etc). We provide the following explanation/interpretation to address this question: first, although MafB is markedly downregulated upon LPS stimulation, it remains to be a process. This duration may allow responses from those MafB dependent/mediated events in WT cells compared to the MafB knockdown/knockout ones. Second, as found in the untreated macrophages, MafB knockdown or knockout already led to the decreased p62 expression. The decreased p62, perhaps together with other MafB regulated mediators, has subjected the MafB knockdown/knockout cells to an augmented activation by the NLRP3 agonists.

Comment 3: Evidence of MafB knockdown is missing in figure 3. It is important to show a representative level of knockdown in these experiments.

Response 3: We have included the suggested data in the revised Figures 3D as well as 3H.

Comment 4: In Figure 6A, it is not explained why a large number of cells with Mitotracker signal have been excluded, as they appear to be positive for both colors but ignored from the analysis.

Response 4: The flow cytometry based mitochondrial damage assay includes two MitoTracker

fluorescent dyes. MitoTracker Deep Red, of which staining is dependent on mitochondrial potential, is brighter in cells with healthy mitochondria than those with damaged ones. MitoTracker Green, which staining of mitochondria is independent on mitochondrial potential, generally indicates the mitochondrial mass in the cells. The gate shows cells with the damaged mitochondria, as reflected by the lower staining of MitoTracker Deep Red. The remaining cells (“double positive”) are generally cells with healthy mitochondria. The percentage is the ratio of the number of the cells with damaged mitochondria to the total number of cells captured (green positive).

Minor comments:

1) In the introduction: ‘with (as for NLRP3 and AIM2) or without (as for NLRP1 and NLRC4) an assistance from the adapter ASC’. NLRP1 and NLRC4 can use ASC if present so this statement is not completely true.

Response 1: We have revised the statement (please see highlighted texts in page 3).

2) In the methods, mtDNA quantification: methodology on subcellular fractionation is missing.

Response 2: We have included the details about mtDNA quantification, and methodology on subcellular fractionation in the revised manuscript.

3) Quantification of ASC specks in 3E would be useful to show how consistent the response is between experimental repeats.

Response 3: The quantification of ASC specks is shown in Figures 3F and 4H.

4) In the discussion: ‘Additionally, we did not find that MafB considerably regulates the transcription of p62 itself (data not shown)’. This data should be included as it provides more mechanistic insight into the author’s proposed mechanism.

Response 4: We have included this data in the revised Figure 7C.

Reviewer #2:

General comments:

The manuscript “MafB inhibits the NLRP3 inflammasome activation by sustaining p62 expression in macrophages” describes a role for the transcription factor MafB as a negative regulator of the NLRP3 inflammasome. The mechanism proposed for this effect involves the expression of p62, which in turn promotes clearance of damaged mitochondria. While the manuscript is of good quality and has good amount of evidence supporting their claims, some points require further elaboration.

Response: Thanks very much for the reviewer’s comments. We have carefully revised the manuscript according to your suggestions.

Comment 1: There are no details regarding immunoprecipitation. Please add in the methods how the immunoprecipitation was performed (lysis buffer and conditions, beads used, antibody dilutions, incubation times, elution method, etc.)

Response 1: We have included detailed information regarding the immunoprecipitation experiment.

Comment 2: Figure 5: Activation of NLRC4 and AIM2 is usually made by transfection of flagellin and poly(dA:dT) respectively. Were these reagents transfected into the cells? If so, how? Additionally, MDP is not a bona fide NLRP1 agonist (Chien-Hsiung Yu et. al., Journal of Molecular Biology, 2018). As it is currently presented, possible effects of MafB on NLRC4, AIM2 and NLRP1 cannot be discarded.

Response 2: Yes, flagellin and poly(dA:dT) were transfected by pre-incubation with LipoJet liposome. We have now included the information in the revised manuscript. As for the second question, we have included a sentence noting the limitation of the experiment (please see highlighted texts in page 14).

Comment 3: Figure 8: While the mitoSOX data on figure 8G is very nice, it is not a direct indicator of mitochondrial damage, as changes in metabolism also affect this assay. The Mitotracker Red/Green assay employed in Figure 6A would be better suited to study mitochondrial damage.

Response 3: Thanks for the reviewer's suggestion. We have performed the suggested experiment. As shown in the revised Figure 8G, LG268 treated macrophages showed diminished mitochondria damage, consistent with the decreased mitochondrial ROS in these cells upon ATP activation.

Comment 4: Claims regarding MafB and p62 need to be softened. For example, the manuscript claims on page 17 "(...) we have firmly established that MafB is required for p62 expression(...)". However, figure 8 data suggests that MafB enhances p62 expression, but it is not required as cells knockdown or knockout for MafB express detectable amounts of p62.

Response 4: We agree with the reviewer. We have toned down the language in the revised manuscript (please see highlighted texts in page 18).

Comment 5: Figure 9 C,D,E: One alternative explanation for these data is that MafB^{-/-} animals have enhanced IL-1b due to increased bacterial burden. Additional data demonstrating that MafB does not impact bacterial clearance in vitro would make this alternative explanation less likely and strengthen the conclusions in this figure.

Response 5: Thanks for the reviewer's insight. We have performed the suggested experiment. As shown in the revised Figure 9G, MafB^{+/+} and MafB^{-/-} macrophages showed similar bactericidal activities on PAK in vitro, suggesting that the impaired PAK clearance in the lungs of Mye MafB^{-/-} mice was likely due to the enhanced NLRP3 inflammasome activation and the worsened lung injury in these mice.

Minor points:

1. Some of the conclusions rely on comparing expression levels using immunoblotting data. While in many cases these differences are visually striking, in some cases the visual differences are too small to draw conclusions. The addition of densitometric data would make these conclusions less ambiguous.

Response 1: We have now provided densitometric data.

2. Some of the text regarding Figure 4 is confusing. On page 11 "(...)the increased IL-1b in the supernatants was due the enhanced inflammasome activation, but not a heightened secretion,

because there was more mature IL-1b in the combined cellular and supernatant extracts from MafB^{-/-} macrophages (Figure 4E).” The presence of more mature IL-1b is usually associated with inflammasome activity, but figures 4B and 4D unambiguously shows that this IL-1b is secreted and found in the supernatants.

Response 2: Our initial intention was to capture the entirety of mature IL-1 β (IL-1 β p17), including that present in the supernatants and that still remaining inside the cells, by combining the cellular and supernatant extracts. After we found that IL-1 β p17 was upregulated in the supernatants, we measured IL-1 β p17 levels in the combined cellular and supernatant extracts and still found the elevated level. These data confirmed that the elevated IL-1 β p17 in the supernatants was resulted from the augmented NLRP3 activation, not due to the cells simply releasing more IL-1 β p17 into the supernatants. We revised to improve the clarity (please see highlighted texts in page 12).

3. Some graphs would benefit from more information on the y-axis. For example, most of Figure 9 would benefit from including what is being quantified in the y-axis (IL-1b pg/mL, etc.).

Response 3: We modified as suggested wherever necessary to improve clarity.

4. Some acronyms are not defined in the manuscript (MAF, KC, ALI) while others are defined multiple times (PAK).

Response 4: Corrected throughout the paper.

5. The English in the manuscript needs some refinement.

Response 5: We refined it to the best of our proficiency.

6. Information regarding LPS concentrations is missing in some figure legends.

Response 6: Added.

7. In the methods section, please add antibody dilutions for the various antibodies used for immunoblots.

Response 7: Now included.

8. Methods: “Determination of ASC homorization”. Is “homorization” correct, or should it be oligomerization? I could not find a definition for the term homorization.

Response 8: Apologize. “homorization” is a typo. It should be “oligomerization”.

Reviewer #3:

General comments:

This article by Cui and colleagues interesting looks to the role of MafB in regulating NLRP3 inflammasome activity. They show, largely using cell culture models, that in the absence of MafB signaling, NLRP3 inflammasome activity is increased demonstrated by enhanced release of active IL-1beta and production of ASC specks.

I found the manuscript to be interesting and relevant for the field, although the following experiments are important to confirm the findings.

Response: We are thankful to the reviewer’s encouragement and suggestions.

Comment 1: None of the western blots are quantified, and the blots should show both the full and cleaved bands for Caspase-1, IL-1b and Gasdermin D

Response 1: We have now provided densitometry quantifications for blots key to the central conclusions. We have shown the full and cleaved bands of these proteins in multiple figures (Figures 3H, 4B, 4E, 5B, 5E, 8A, etc.).

Comment 2: Some western blots do not show the groups together on the same gel e.g. Fig 1F, Fig3D, Fig 8A

Response 2: The bands were from the same gel. They were not grouped together only because unrelated data to the paper in between were not included. Nonetheless, we repeated the experiments in Fig. 1F and Fig. 3D and show them in the same group. Data in Fig. 8A were intended to be compared within each activation group and thus being presented as such for easier interpretation.

Comment 3: A number of experiments are missing the control, untreated groups, e.g. Fig 2B, 2D, 3A-C, 3G, 4A-D, 4I-J, Fig8A

Response 3: The missing controls in the referenced figures either had an undetectable level or were set as 1 fold to calculate the fold change in the treatment groups. They were not included to make the figure less busy and easier to go through.

Comment 4: Fig 3E is not quantified

Response 4: We have provided the quantification.

Comment 5: Labelling is inconsistent e.g. sometimes the graph states LPS+ATP other times just ATP (e.g. Fig 6C-E). The graph sizes also vary within the same figures

Response 5: We apologize for the confusion. We have modified the figures to improve the clarity.

Comment 6: For Fig 7, the authors state that “p62 knockdown diminished the NLRP3 inflammasome activation...” (page 14) however in Fig 7E, the levels of IL-1b are actually significantly increased after p62 siRNA treatment, and the bands on the westerns on 7F also looks stronger. Can the authors comment on this?

Response 6: We apologize. It was a mistake that we have now corrected. “p62 knockdown diminished the NLRP3 inflammasome activation...” should read as “p62 knockdown augmented the NLRP3 inflammasome activation...”.

Comment 7: If ubiquitination is as necessary as the authors suggest, an experiment inhibiting ubiquitination would be helpful for the manuscript. The westerns blots provided in Fig 1K are hard to interpret

Response 7: We appreciate the reviewer’s insight. Because we have not identified the MafB E3 ubiquitin ligase(s), we cannot specifically block MafB ubiquitination. However, as an alternative, in a new experiment, we blocked the E1 ubiquitin-activating enzyme with the inhibitor MLN7243 and found that the E1 inhibitor led to MafB accumulation and diminished LPS induced MafB degradation (Figure 1K). These data further support the notion that MafB undergoes rapid ubiquitin mediated turnover when LPS signaling inhibits MafB transcription.

Comment 8: It is surprising the authors did not make use of western blots for their in vivo experiment in Fig 9. I suggest to add those as without confirming the NLRP3 inflammasome components, the IL-1beta could be coming from other pathways

Response 8: We appreciate the suggestion. We performed additional experiments and found that the NLRP3 activation was enhanced in the lungs of LPS treated MafB^{-/-} mice (Figure 9C).

Comment 9: There are some typos throughout e.g. p12 “Gasdermine” and “oilgomerization” I suggest going through the manuscript in detail with a spell checker

Response 9: We apologize for the typos. We have made the best effort to correct the similar errors.

Comment 10: There is a recent paper from Billingham et al 2022 that shows mitochondrial ROS is not necessary for NLRP3 activation. This finding, and others showing mitochondria-independent mechanisms for NLRP3 inflammasome activation should be detailed more in the introduction, which currently reads that basically all NLRP3 inflammasome triggers work by inducing mitochondrial disruption.

Response 10: We thank the reviewer’s insight. We have includes sentences to reference these studies in the revised introduction (please see highlighted texts in page 4).

Comment 11: In the discussion the authors write “MafB is an important factor in both the priming and activating phases during the NLRP3 inflammasome activation”. They do show that MafB has a role in the NLRP3 assembly (i.e. increased IL-1beta and production of ASC specks), but how it may affect NLRP3 priming is not clear. Indeed they also comment that “MafB specifically inhibiting the NLRP3 inflammasome”. This seems not accurate, as if it was true inhibition, MafB overexpression would block NLRP3 activation. Rather, MafB seems to be regulating, or fine tuning NLRP3 inflammasome activity, and I suggest updating the text to reflect that.

Response 11: We thank the reviewer’s suggestion. We have revised these sentences to acknowledge the limitation of the study and to reflect more precisely the implication of findings (please see highlighted texts in page 20).

Reviewers' comments:

Reviewer #1 (Remarks to the Author):

The authors have addressed my concerns and as such the manuscript is now recommended for publication.

Reviewer #2 (Remarks to the Author):

The authors addressed all my comments, and I am satisfied with the additional data provided and other corrections to the original manuscript.

Reviewer #3 (Remarks to the Author):

The authors have clarified a number of points that I raised during the first review round, and the changes that they incorporated are easy to follow in the undated manuscript. Unfortunately, a number of the key suggestions were not fully addressed, which I detail below. These points are central to the interpretation of the manuscript, without these details a number of the points raised by the authors are hard to interpret.

1. In the first round I suggested that Cui and colleagues provide quantification for the western blots, and that the blots should show both the full and cleaved bands for Caspase-1, IL-1b and Gasdermin D. I am surprised to read their response which states "We have now provided densitometry quantifications for blots key to the central conclusions. We have shown the full and cleaved bands of these proteins in multiple figures (Figures 3H, 4B, 4E, 5B, 5E, 8A, etc.)." On looking through these figures, 3H is the only one to show on the same blot the full length and cleaved caspase-1 and GSDMD. In all other figures, that is not the case. Showing the full length and cleaved western blot for IL-1beta, caspase-1 and GSDMD in the lysate and IL-1beta supernatant is central to support the claims on NLRP3 inflammasome activation. These important blots are missing from Fig 1(A, B, C and F), Fig 2(A, F), Fig 3(D and H), Fig 4(B, D, E, I, J, K), Fig 5(B, E), Fig 6(G), Fig 7(B, D, G), Fig 9(C), Fig S1(A and E), Fig S3(D and E). The curious thing is that the authors surely have this data, the antibodies (especially for caspase-1 and IL-1b) pick up the pro and cleaved forms, therefore this information will all be in the western blot images that they cropped for the publication. I am also missing the updated quantification. It appears only Fig 9C has western blot quantification in a graph form, and even that is only partial showing caspase-1 p20 and gasdermin D p30. Some of the other figures have the intensity values added above/below the western blot image, but that is incredibly hard to interpret. This should also be in a graph format with statistics to allow proper interpretation of the data.

2. In the previous round I pointed out that a number of experiments are missing the control, untreated groups, e.g. Fig 2B, 2D, 3A-C, 3G, 4A-D, 4I-J, Fig8A. The authors reply "The missing controls in the referenced figures either had an undetectable level or were set as 1 fold to calculate the fold change in the treatment groups. They were not included to make the figure less busy and easier to go through." Controls are a crucial part of every experiment and necessary for the reader to interpret the actions of the stimuli used. As the authors have this data, I strongly suggest including it into the manuscript.

3. A previous comment was in relation to the discussion, where the authors write "MafB is an important factor in both the priming and activating phases during the NLRP3 inflammasome activation". They do show that MafB has a role in the NLRP3 assembly (i.e. increased IL-1beta and production of ASC specks), but how it may affect NLRP3 priming is not clear, therefore the "priming"

part should be removed from that sentence. The authors state the following on page 11 "we knocked down MafB in PMs and BMDMs and treated the cells with LPS, and found that knockdown of MafB has no effect on LPS induced NLRP3 or pro-IL-1 β , at either the protein or the transcriptional level". That would even suggest MafB has no effect on NLRP3 priming. Indeed, they also comment that "MafB specifically inhibiting the NLRP3 inflammasome". This is not accurate, as if it was true inhibition, MafB overexpression would block NLRP3 activation. Rather, MafB seems to be regulating, or fine tuning NLRP3 inflammasome activity, and I suggested updating the text to reflect that. In the abstract the authors state that "MafB is an important negative regulator of the NLRP3 inflammasome". This sentence is the most reflective of the data, and a more appropriate statement than "inhibition", which is used frequently throughout the manuscript.

I understand that the authors have added a text on the project limitations at the end of their discussion, but these key sentences remain unchanged. As the sentences do not reflect the content of the manuscript, it is important that they are toned down.

Reviewer #3 (Remarks to the Author):

General comments. The authors have clarified a number of points that I raised during the first review round, and the changes that they incorporated are easy to follow in the updated manuscript. Unfortunately, a number of the key suggestions were not fully addressed, which I detail below. These points are central to the interpretation of the manuscript, without these details a number of the points raised by the authors are hard to interpret.

Response: We genuinely appreciate Reviewer 3's thorough feedback and the dedication to upholding rigorous data quality standards – values we share deeply in our research pursuits.

While we understand Reviewer 3's reservations about certain aspects of our research, we believe that some of the concerns arise from potential misunderstandings about our data flow and acquisition process. We take full responsibility for any ambiguities and will strive to clarify our methods and rationale further below.

Comment 1: In the first round I suggested that Cui and colleagues provide quantification for the western blots, and that the blots should show both the full and cleaved bands for Caspase-1, IL-1b and Gasdermin D. I am surprised to read their response which states “We have now provided densitometry quantifications for blots key to the central conclusions. We have shown the full and cleaved bands of these proteins in multiple figures (Figures 3H, 4B, 4E, 5B, 5E, 8A, etc.)” On looking through these figures, 3H is the only one to show on the same blot the full length and cleaved caspase-1 and GSDMD. In all other figures, that is not the case. Showing the full length and cleaved western blot for IL-1beta, caspase-1 and GSDMD in the lysate and IL-1beta supernatant is central to support the claims on NLRP3 inflammasome activation. These important blots are missing from Fig 1(A, B, C and F), Fig 2(A, F), Fig 3(D and H), Fig 4(B, D, E, I, J, K), Fig 5(B, E), Fig 6(G), Fig 7(B, D, G), Fig 9(C), Fig S1(A and E), Fig S3(D and E). The curious thing is that the authors surely have this data, the antibodies (especially for caspase-1 and IL-1b) pick up the pro and cleaved forms, therefore this information will all be in the western blot images that they cropped for the publication.

I am also missing the updated quantification. It appears only Fig 9C has western blot quantification in a graph form, and even that is only partial showing caspase-1 p20 and gasdermin D p30. Some of the other figures have the intensity values added above/below the western blot image, but that is incredibly hard to interpret. This should also be in a graph format with statistics to allow proper interpretation of the data.

Response 1: In our initial studies, once it was established that MafB knockdown or knockout did not affect the expression of pro-IL-1 β and NLRP3 in either untreated or LPS-treated macrophages, our primary focus shifted to understanding MafB's role in regulating NLRP3 inflammasome activation. As is well-understood in this field, the primary indicators for assessing NLRP3 activation are ELISA, which determines the level of mature IL-1 β in the supernatants of LPS primed cells treated with a NLRP3 activator (like ATP and Nigericin), and Western blot, which pinpoints the levels of cleaved forms of IL-1 β , Caspase 1, and Gasdermin D in the precipitated proteins from the supernatants or combined cell and supernatant extracts.

Our standard practice did not involve blotting for pro-IL-1 β , pro-Caspase 1, or total Gasdermin D when the sole aim was to gauge the level of NLRP3 activation. This is primarily because the cleaved forms of these proteins - not the uncleaved ones - are indicative of NLRP3 activation. Consequently, we often cut the membrane at 37KD and subsequently blotted the lower portion of the membrane for the cleaved proteins while blotting the upper membrane for MafB, NLRP3, p62, etc. However, for clarity, we've revisited this approach in Figure 3D to showcase both the uncleaved and cleaved forms of Casp1 and GSDMD.

Furthermore, it's essential to acknowledge that while Western blot is indeed a powerful research tool, its qualitative or semi-quantitative nature inherently limits its application for rigorous statistical comparisons across distinct blots performed at varying times. This limitation is not unique to our research but is universally acknowledged in many esteemed publications, including those in the Nature series. For this manuscript, our approach has been to depict the quantification of blots sourced from the same samples in a representative experiment, with numerical values expressing fold changes relative to control groups (e.g., con si, +/+, etc.), which are considered as a 1-fold reference..

Comment 2. In the previous round I pointed out that a number of experiments are missing the control, untreated groups, e.g. Fig 2B, 2D, 3A-C, 3G, 4A-D, 4I-J, Fig8A. The authors reply “The missing controls in the referenced figures either had an undetectable level or were set as 1 fold to calculate the fold change in the treatment groups. They were not included to make the figure less busy and easier to go through.” Controls are a crucial part of every experiment and necessary for the reader to interpret the actions of the stimuli used. As the authors have this data, I strongly suggest including it into the manuscript.

Response 2: Our previous research consistently indicated minimal pro-IL-1 β expression in naïve macrophages. Additionally, in LPS treated macrophages that weren't stimulated by a NLRP3 activator, we observed virtually no cleavage of pro-IL-1 β , pro-Caspase 1, or Gasdermin D, regardless of the MafB status. This is the primary reason we often did not measure the levels of pro-IL-1 β and other proinflammatory cytokines in naïve macrophages or the cleaved form of these proteins in the absence of an inflammasome activator stimulus.

However, recognizing the importance of controls for comprehensive understanding, we have now included them in the revised figures wherever the data were available. You can refer to the updated figures, specifically Fig. 2B, 2C, 2D, 2H, 3A, 3B, 3E-F, 3G, 4A-C, 4G-H, 4I-J, and 8A, in the revised manuscript.

Comment 3: A previous comment was in relation to the discussion, where the authors write “MafB is an important factor in both the priming and activating phases during the NLRP3 inflammasome activation”. They do show that MafB has a role in the NLRP3 assembly (i.e. increased IL-1beta and production of ASC specks), but how it may affect NLRP3 priming is not clear, therefore the “priming” part should be removed from that sentence. The authors state the following on page 11 “we knocked down MafB in PMs and BMDMs and treated the cells with LPS, and found that knockdown of MafB has no effect on LPS induced NLRP3 or pro-IL-1 β , at either the protein or the transcriptional level”. That would even suggest MafB has no effect on NLRP3 priming. Indeed, they also comment that “MafB specifically inhibiting the NLRP3

inflammasome”. This is not accurate, as if it was true inhibition, MafB overexpression would block NLRP3 activation. Rather, MafB seems to be regulating, or fine tuning NLRP3 inflammasome activity, and I suggested updating the text to reflect that. In the abstract the authors state that “MafB is an important negative regulator of the NLRP3 inflammasome”. This sentence is the most reflective of the data, and a more appropriate statement than “inhibition”, which is used frequently throughout the manuscript.

I understand that the authors have added a text on the project limitations at the end of their discussion, but these key sentences remain unchanged. As the sentences do not reflect the content of the manuscript, it is important that they are toned down.

Response 3: We appreciate your feedback and have made the necessary revisions to better reflect the data and findings. Specifically, we have amended the phrasing from "inhibit" to "regulate" and adjusted the statement, “MafB is an important factor in both the priming and activating phases during the NLRP3 inflammasome activation”, throughout the manuscript as suggested. Please refer to the highlighted text in the revised version to review these changes.